# Evaluation of the Influence of Disturbances on Forest Vegetation Using the Time Series of Landsat Data: A Comparison Study of the Low Tatras and Sumava National Parks

**Premysl Stych** [1] **, Josef Lastovicka** [1,*]**, Radovan Hladky** [1,2] **and Daniel Paluba** [1]

[1]  Department of Applied Geoinformatics and Cartography, Faculty of Science, Charles University, Albertov 6, 128 43 Prague 2, Czech Republic; stych@natur.cuni.cz (P.S.); rado.hladky@gmail.com (R.H.); daniel.paluba@natur.cuni.cz (D.P.)

[2]  Administration of the Low Tatras National Park, State Nature Conservation of the Slovak Republic (NAPANT), Partizanska cesta 69, 974 01 Banska Bystrica, Slovakia

*  Correspondence: lastovj1@natur.cuni.cz

**Abstract:** This study focused on the evaluation of forest vegetation changes from 1992 to 2015 in the Low Tatras National Park (NAPANT) in Slovakia and the Sumava National Park in Czechia using a time series (TS) of Landsat images. The study area was damaged by wind and bark beetle calamities, which strongly influenced the health state of the forest vegetation at the end of the 20th and beginning of the 21st century. The analysis of the time series was based on the ten selected vegetation indices in different types of localities selected according to the type of forest disturbances. The Landsat data CDR (Climate Data Record/Level 2) was normalized using the PIF (Pseudo-Invariant Features) method and the results of the Time Series were validated by in-situ data. The results confirmed the high relevance of the vegetation indices based on the SWIR bands (e.g., NDMI) for the purpose of evaluating the individual stages of the disturbance (especially the bark beetle calamity). Usage of the normalized Landsat data Climate Data Record (CDR/Level 2) in the research of long-term forest vegetation changes has a high relevance and perspective due to the free availability of the corrected data.

**Keywords:** time series; Landsat; vegetation indices; the Low Tatras; bark beetle disturbance; Sumava

---

## 1. Introduction

The issue of the time series has been a highly discussed subject recently, particularly in the field of observations of local and global change. The time series from Landsat gives us insights from the 1980s that can be used for a variety of analyses. This is the only continuous mission in the field of high-resolution data. This fact brings many problems that need to be addressed through the pre-processing of data: from the non-homogeneity of the sensed environment, through the different types of sensors with different radiometric and spectral resolutions, to the aging of the sensors. Several possible methods of processing the long time series are currently being used. Due to the low temporal resolution (for example, unlike Sentinel-2) of the Landsat images, it is necessary to pre-process the data for a possible quantitative comparison. We can use, for example, spectral unmixing methods [1], fractional methods [2], radiometric normalization and linear image regression [3–13]. Other methods such as cross-calibration methods that combine sets of multiple data types [14] are also used.

Thanks to the correct pre-processing of the raw data, it is possible to monitor several local and global phenomena. From the local phenomena, changes in forest ecosystems have been the most observed. In forest ecosystems, it is possible to observe the decrease of forests in tropical areas, the cultivation of palm trees or the range of biotic and abiotic disasters, among which bark beetle calamities are associated, which is also pertinent to this article. The massive expansion of bark beetle calamities was mainly due to the planting of non-indigenous species of coniferous forests in Central Europe. In the last few years, there have been large droughts that evoke dissemination of the pests to damage the weakened vegetation [15] and also allows the pests to dig out several times in a year. The second major factor is global change and global warming, which helps to exacerbate calamities.

Current studies [16] help us determine the parameters of the environment into which the pests are spreading. The quality of the trees, altitude, soil quality, groundwater content, or aspect belong among the factors that affect dissemination. Besides medium multispectral data, hyperspectral data [17] or very high-resolution multispectral data [18] are also used. These data help us determine the individual phases of damage more accurately, such as the Red Attack phase. High-resolution multispectral data help us locate the areas by various indicators and detect the particular phase or quantity of damage [19–21]. For medium resolution, only the wider changes [22–25] can be localized. On this basis, this study documented the process of bark calamities in various phases in the selected localities of two national parks using a time series of satellite imagery that was pre-processed with relative radiometric normalization for the purpose of comparison of values over time. As an indicator of change, specific types of vegetation indices were selected.

This work was focused on evaluating of changes in the forest areas in the selected localities of the Low Tatras NP (National Park) and Sumava NP using TS methods. The creation of the TS is based on the Landsat images. The changes in the forest vegetation induced by various disturbance processes were evaluated based on the selected vegetation indices, their tentative capabilities being tested and validated by comparing the in-situ data. An important part of this study was the evaluation of the data, methods and results achieved for the landscape and nature management and nature conservation institutions.

The main objectives of this work were:

- To evaluate and compare changes in the forest areas in the selected localities of the Low Tatras National Park and Sumava National Park using TS methods, which use normalized Landsat data
- To evaluate the suitability of individual vegetation indices for the detection of different types of biotic and abiotic disturbances
- To validate and interpret the results using in-situ data
- To discuss and recommend the suitability of the Earth Observation for nature conservation and management of the Low Tatras National Park and Sumava National Park.

*Observed Area*

The Low Tatras National Park (in Slovakia) is 73 km$^2$. The dominant land cover is forest; thus, we can find extensive forest ecosystems. In many parts of the Low Tatras, spruce forests dominate. Their development takes place under the influence of natural and anthropogenic factors. The integrity of the forest ecosystems has been disturbed by several abiotic events (wind, snow, frost, and avalanches), biotic agents (sub-insects) or anthropogenic influences (e.g., air pollution).

On 19 November 2004, from 15:00 to 24:00, a windstorm passed through the area of the Low Tatras with a maximum speed of about 175 km/h. The territory of the Low Tatras National Park and its protection zone was damaged and extensive forest stands in several areas and localities were destroyed. The wind storm in November 2004 hit the territory of NAPANT very strongly and, as a result, led to the reproduction of the European bark beetle (Ips typographus). During the summer of 2007, extremely favorable conditions for grafting of subcortical insects were created in almost all non-native and natural spruces in Slovakia (Figure 1). The observed sites of this study are located in the Dumbier part of the Low Tatras (Western Carpathians), in the Mlynna valley. It is in an about

7 km long valley. After the wind calamity in 2004, the valley area was affected by a biotic bark beetle calamity, mainly culminating in 2009 (mass pine devastation). The sites of interest were categorized according to the type of disturbance. The first site represents the territory with the wind disturbance in 2004, the second represents the prevailing beetle disturbance between 2006 and 2009, the third site has minimal disturbance, the fourth site represents a place with the beetle disturbance (2005) immediately after wind calamity in 2004 and the last site represents a place with minimal disturbance (Figure 2 and Table 1).

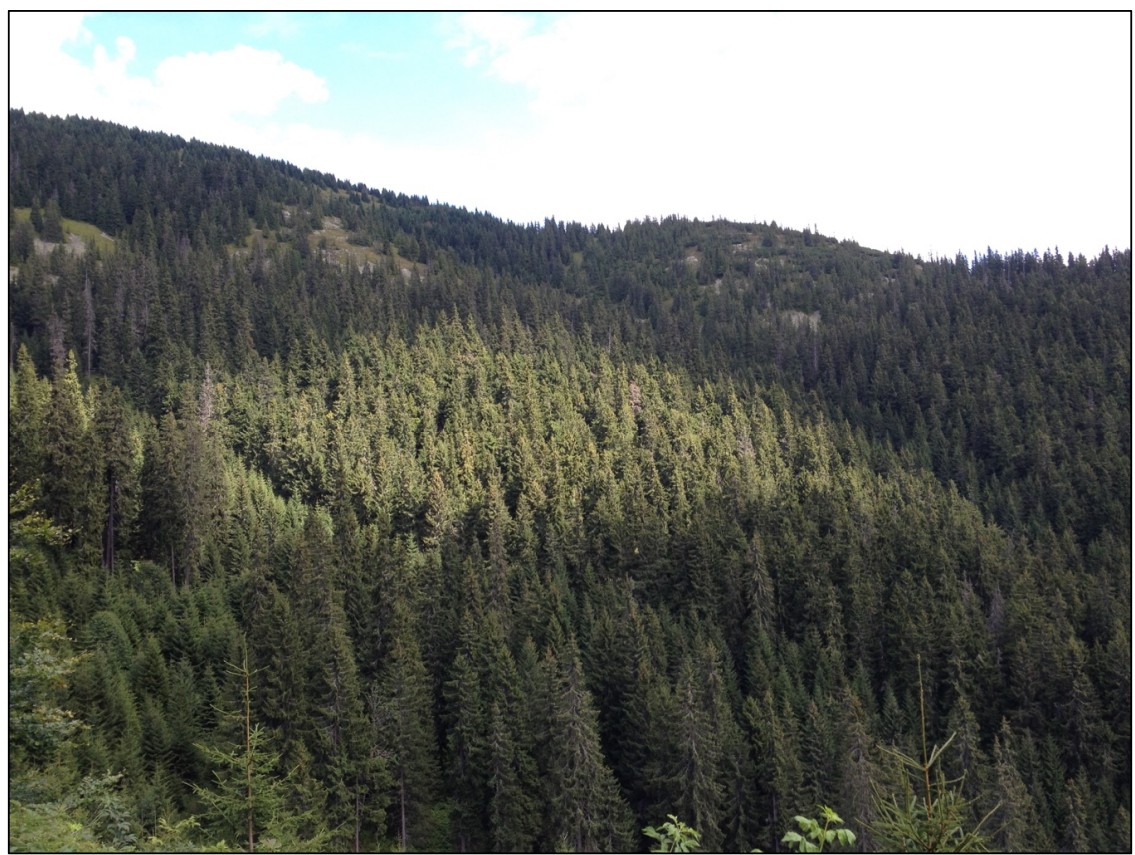

**Figure 1.** The first phase of a bark beetle calamity in the Low Tatras (Photo by J. Lastovicka).

**Table 1.** Overview of the selected localities in the Low Tatras NP with the GPS position in WGS 84.

| ID | Type of Disturbance | Year | Northing | Easting |
|----|---------------------|------|----------|---------|
| 1 | Wind calamity | 2004 | 48.9047556 | 19.6364333 |
| 2 | Bark beetle calamity | 2006–2009 | 48.9155019 | 19.6520547 |
| 3 | Minimal disturbance | – | 48.9010178 | 19.6532992 |
| 4 | Bark beetle with wind calamity | 2004–2005 | 48.903718 | 19.716530 |
| 5 | Minimal disturbance | – | 49.0019478 | 19.6275692 |

The Sumava is an extensive mountain range on the border of Czechia, Austria and Germany (Bavaria). The Sumava (Bohemian) Forest together with the neighboring Bavarian Forest creates the most extensive forest landscape in Central Europe, called the "Green Roof of Europe". The area of the park is more than 900 km². The Sumava forests are dominated by spruce.

During the 1990s in the last century, a bark beetle calamity occurred in the Czech part of Sumava NP with the culmination during 1995–2001. The Kyrill gales had a strong impact on the forest vegetation in 2007. In 2008, there was a sharp rise in the bark beetle affected forest areas and it continued to grow until 2010. In the period from 2011 to 2014, there was a slight decline in the intensity of the disturbance with a stabilization period after 2014 (Figure 3). The first site of interest in this study

represents the territory with the wind and bark beetle disturbance, the second and fourth represent the bark beetle disturbance, and the third and fifth site are with minor disturbances (Figure 4 and Table 2).

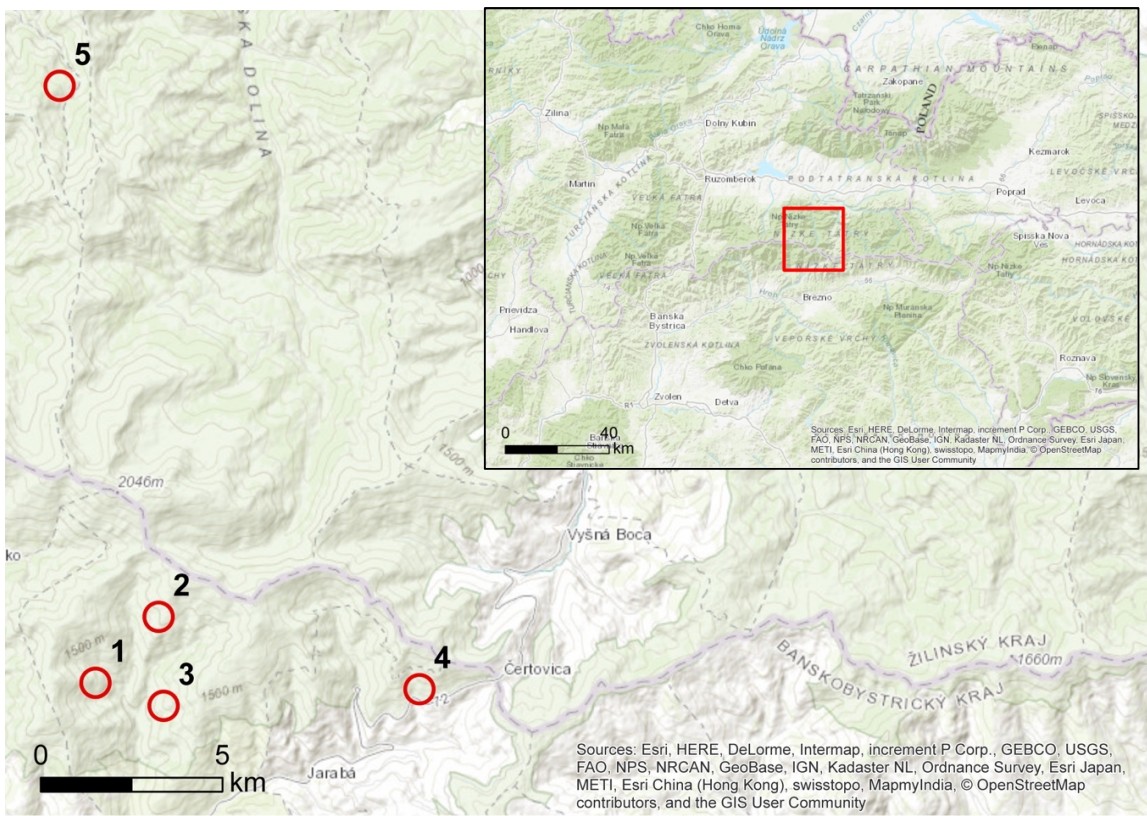

**Figure 2.** Map of the used localities in the Low Tatras (Source: Own work/ESRI ArcMap Basemap). The number of the points represents the ID numbers.

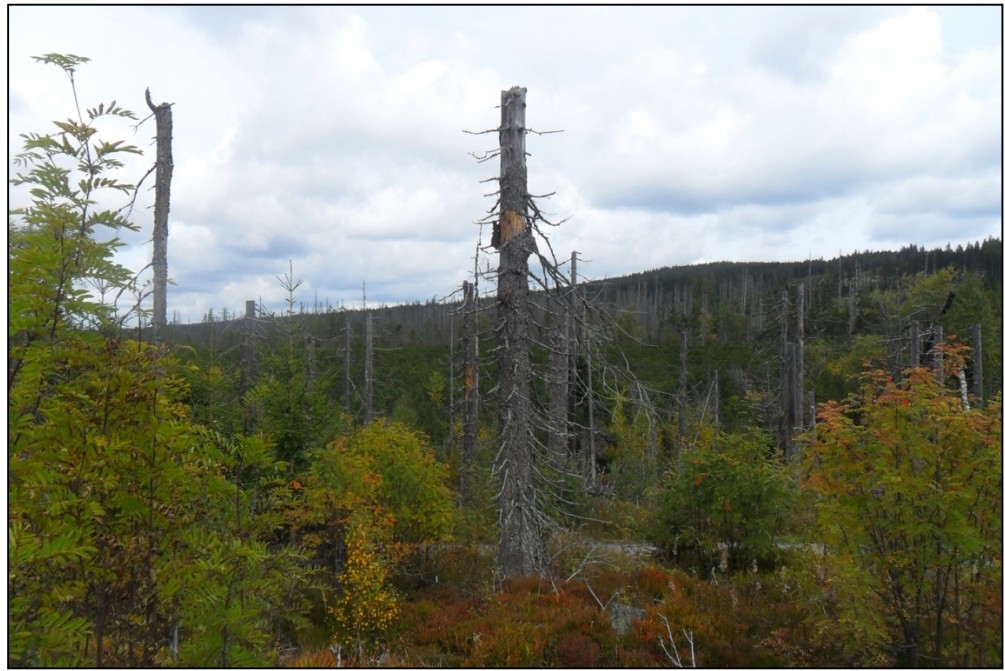

**Figure 3.** A forest affected by a bark beetle and wind disturbance in the Sumava NP (source: R. Hladky).

**Table 2.** Overview of the selected localities in the Sumava NP with the GPS position in WGS 84.

| ID | Type of Disturbance | Year | Northing | Easting |
|----|---------------------|------|----------|---------|
| 6 | Bark beetle calamity with wind and non-natural recovery | since 2007 | 48.973628 | 13.561746 |
| 7 | Bark beetle calamity and natural recovery | since 2008 | 48.983655 | 13.561720 |
| 8 | Minimal disturbance | – | 48.989884 | 13.558225 |
| 9 | Bark Beetle with natural recovery | since 2009 | 48.9847661 | 13.5227772 |
| 10 | Minimal disturbance | – | 49.0451933 | 13.4761361 |

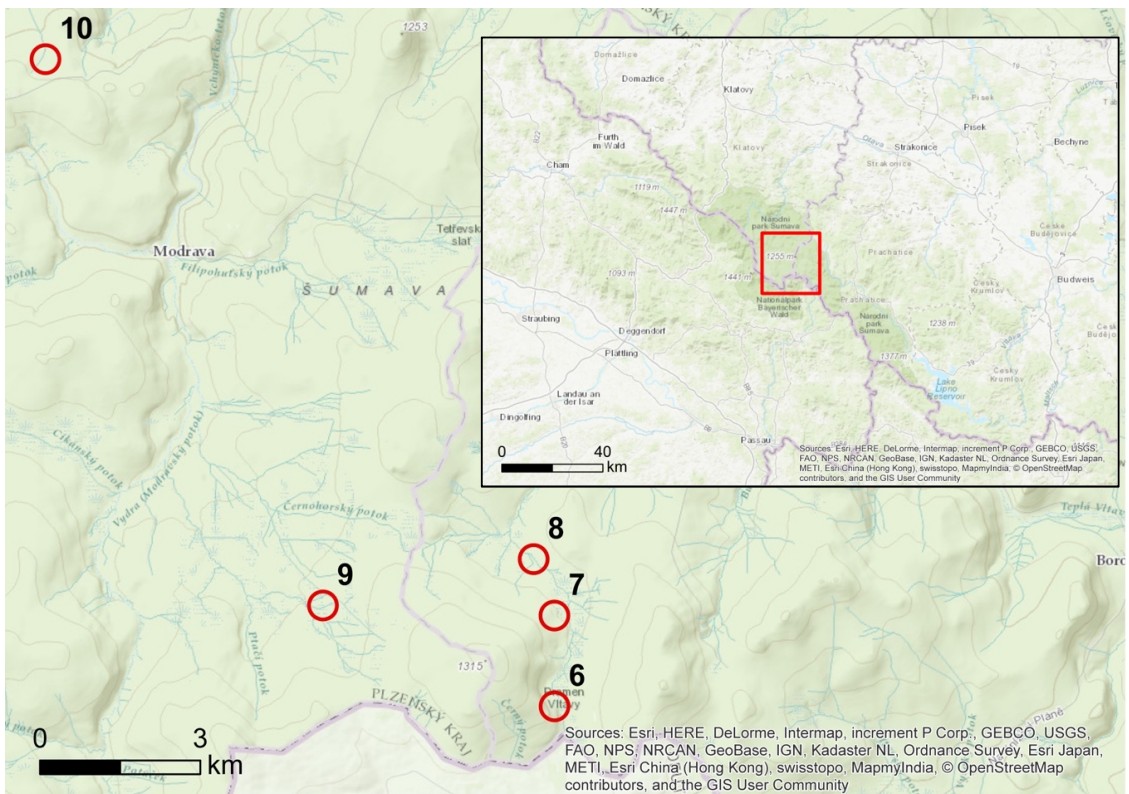

**Figure 4.** Map of the used localities in the Sumava NP (Source: Own work/ESRI ArcMap Basemap). The number of the points represents the ID numbers.

## 2. Materials and Methods

### 2.1. Data

Satellite and in-situ data were used to process the study. Landsat Climate Data Record (CDR) satellite imagery was used as the remote sensing data. The distributed Landsat CDR data were pre-processed with geometric and atmospheric corrections using the CDR database. For more information on the methodologies, see, e.g., [26,27].

A sufficient number of usable Landsat satellite images (data from the Landsat 4–8 missions) were found for our areas of interest. Eleven images were finally selected for each of the Low Tatras NP (Table 3) and the Sumava NP (Table 4). A crucial condition for the selection of the data was their minimum cloud cover and no significant haze infection. Another important aspect of the selection was the date of the acquisition. According to many authors (e.g., Vogelmann et al. [28] and Griffiths et al. [29,30]), the summer to the first autumn phase is an appropriate time for the assessment of the forest vegetation. Based on these works, it was decided to select data in the months of July to September.

**Table 3.** Overview of the used Landsat images in the Low Tatras NP.

| Name | Date | Sensor |
|---|---|---|
| LC81880262013219LGN00 | 7 August 2013 | Landsat 8 |
| LC81880262015193LGN00 | 12 July 2015 | Landsat 8 |
| LE71880261999221SGS01 | 9 August 1999 | Landsat 7 |
| LE71880262001242SGS00 | 30 August 2001 | Landsat 7 |
| LT51880261994183XXX02 | 2 July 1994 | Landsat 5 |
| LT51880262005245KIS00 | 2 September 2005 | Landsat 5 |
| LT51880262006200KIS01 | 19 July 2006 | Landsat 5 |
| LT51880262007203MOR00 | 22 July 2007 | Landsat 5 |
| LT51880262009240KIS00 | 28 August 2009 | Landsat 5 |
| LT51880262011198MOR00 | 17 July 2011 | Landsat 5 |
| LT41880261992202XXX02 | 20 July 1992 | Landsat 4 |

The in-situ data, used for the validation of the results from the Landsat data analysis, were obtained from the administration of the national parks and from their own field survey. The information obtained from the national parks included the field records of foresters, information from forest management plans and archive records of nature conservation documentation. The forest management plan is legislatively enshrined in the Forest Law on Forest Economic Planning and it provided data on the status of the forests and their past management: the area and category of the forest, the age of the stock and the age, stocking and representation of the individual trees, the average height, stockpile and method of management and proposal of the economic measures.

**Table 4.** Overview of the used Landsat images in the Sumava NP.

| Name | Date | Sensor |
|---|---|---|
| LC81920262013215-SC20160702161259 | 3 August 2013 | Landsat 8 |
| LC81920262015221-SC20160702161912 | 9 August 2015 | Landsat 8 |
| LE71920262002209-SC20160702153836 | 28 July 2002 | Landsat 7 |
| LT51920261994211-SC20160702152112 | 30 July 1994 | Landsat 5 |
| LT51920261998222-SC20160702151843 | 10 August 1998 | Landsat 5 |
| LT51920262004223-SC20160702151849 | 10 August 2004 | Landsat 5 |
| LT51920262005241-SC20160702151622 | 29 August 2005 | Landsat 5 |
| LT51920262006196-SC20160702151531 | 15 July 2006 | Landsat 5 |
| LT51920262007231-SC20160702151832 | 19 August 2007 | Landsat 5 |
| LT51920262009236-SC20160702151537 | 24 August 2009 | Landsat 5 |
| LT51920262010191-SC20160702151637 | 10 July 2010 | Landsat 5 |

*2.2. Data Processing*

To determine the state and changes in the vegetation from the satellite images, the key information was placed on the spectral characteristics of the vegetation species studied. Each species of vegetation has specific characteristics, and we can detect them and determine their properties (e.g., health status) based on this knowledge. Spectral reflection can be represented by a curve that can be divided into three basic parts according to the main structural properties: the area of pigmentation absorption, the cellular structures and the area of water absorption [31]. The time series analyses monitor the changes in the vegetation over long periods of time. The spectral characteristics of studied objects are often used for estimating the health phase of vegetation or it could be based on the spectral characteristics of the derived data, such as the vegetation indices.

To ensure compatibility, the Landsat data source was used. As mentioned above, the atmospheric and radiometric corrections did not need to be performed on the CDR data because the images are distributed with the correction and conversion to surface reflectance (i.e., surface reflectance). The Fmask [32] feature was used to create cloud-free images and to mask the clouds, shadows, snow and other disturbing elements.

It is clear from many studies that a significant problem in creating the time series is to ensure compatibility between the types of data that are often taken by multiple types of sensors [33]. A different sensor type and a different acquisition time may have a large influence on the result of the time series analyses. For this reason, relative radiometric normalization was used to eliminate the influence of the different acquisition times, the different phenological phases of the vegetation and the influence of the different spectral and radiometric characteristics of the different sensors. Thanks to normalization, we can reduce the effects of the different time and place of acquisition, the different positions of the sun, and the different radiometric and spectral differences.

Based on previous empirical testing, the PIF (Pseudo-Invariant Features) Linear Based method was selected for the normalization purposes. This normalization was selected based on the recommendations of many studies [33,34] and the testing of relative radiometric normalizations in the article by Lastovicka et al. (2017) [35]. It is a relatively standard proven method, which already has its origins in the 1990s, and the PIF Linear Based method is probably the most commonly used [34]. There are many variations of PIF and it is used in many programs for time series (IR-MAD/MAD CAL algorithm). The linear-based method is based on linear alignment using linear regression. For the PIF linear-based method, a classical linear regression ($y = b_1 * x + b_2 + \varepsilon$) was used, which represents the approximation of the known values using the smallest square method.

Thus, by linear regression, we looked for the most suitable parameters, $b_1$ and $b_2$, to approximate the two observed frames in the correlation diagram. The values $b_1$ and $b_2$ represent the slope and intercept, respectively [34].

The method consists of finding the pseudo-invariant objects and the invariant objects in two observed images that occur as clusters [33,36–38]. Using a scatter plot, the values of both images and the search for the unmodified objects were displayed manually or automatically using the algorithm [39]. The invariant areas required for this method were selected based on the NDVI index, where the minimum area of the index was the appropriate area. Roads, parking areas, roofs and other long-standing elements were among the selected invariant objects. These selected elements are suitable for normalization calculations. Then, the regression parameters were computed [33]. The equalization was performed for each spectrum band separately—the band-by-band method [39].

For the data normalization of the CDR Landsat data, custom applications were developed in the MATLAB environment. The developed application was inspired by the TimeSync web application (made by Oregon State University—http://timesync.forestry.oregonstate.edu [40]), which has limitations in accessing custom data as well as in bulk data over the Internet. Our application allows one to work with any type of satellite data in an off-line mode. Thanks to the relatively standardized MATLAB encryption, this application is transferable to a wide range of software and operating systems. The application itself consists of several algorithms for which two user interfaces were created. The first user interface allows seeing the differences between the two spectral bands that are needed for normalization purposes. The scatter graph and interdependence were displayed using a linear regression curve.

After the data standardization process, the selected vegetation indices were calculated, and the time series charts were created. To create the curves, the developed applications were used. For the specified location, the application created time series charts. The calculations take the pixel values of the site or its surroundings ($3 \times 3$ pixels) into account. In this study's case, the closest neighborhood of the site was used ($3 \times 3$ pixels). Nine pixels were selected for one pixel in the neighborhood of the central pixel (the localities were chosen from a larger area of disturbed or non-disturbed places, thus the values were averaged for the approximate value errors or outlying values). For a more detailed description of this application, see the work of Lastovicka et al. [35].

Based on several elaborated studies [34,41–44], it was decided to use the list of vegetation indices shown in Table 5.

**Table 5.** List of the vegetation indices and their expressions.

| Vegetation Index | Shortcut | Equation |
|---|---|---|
| Foliar Moisture Index | FMI (value * 10) | (NIR)/(RED * SWIR) |
| Normalized Difference Moisture Index | NDMI | (NIR − SWIR)/(NIR + SWIR) |
| Normalized Difference Vegetation Index | NDVI | (NIR − RED)/(NIR + RED) |
| Simple Ratio Index | SR (value/100) | (NIR)/(RED) |
| Transformed Vegetation Index | TVI | sqrt ((NIR-RED)/(NIR + RED) + 0.5) |
| Wide-band Normalized Difference Infrared Index | wNDII | (2 * NIR − SWIR)/(2 * NIR + SWIR) |

Note: The FMI and SR values were optimized for the value level to be comparable with the other chosen indices in the graphs.

The masking and calculation of the vegetation indices was performed in ENVI 5.3. The final database created for the time series analysis included calculations of six vegetation indices from the Landsat CDR normalized data by the PIF Linear Based method.

For the statistical approach, the minimum (MIN), the maximum (MAX), the difference between maximum and minimum (MAX-MIN) and the standard deviation (St. Deviation) were calculated.

For both areas of interest, the aggregate mathematical statistical characteristics were calculated. From the locations where calamities occurred, the average values for the given period were calculated and the graphs for all the interest areas were plotted.

## 3. Results

This section focuses on the description and interpretation of the value of the vegetation indices in the 10 different localities in the Low Tatras and Sumava National Parks that have been under different development influencing the forest areas (damage from wind calamity, bark beetle and forest vegetation without any significant influence). The results describe and interpret the values of the studied vegetation indices during the observed period (1992–2015) and evaluate their suitability for the assessment of the forest changes. The results should indicate the applicability of the different vegetation indices or their combinations for the detection of the different types of disturbances.

### 3.1. Locality 1

A significant part of the Low Tatras National Park was damaged by the wind calamity of Elizabeth, which took place on 19 November 2004. The first site represents the area heavily damaged by this event, followed by the natural regeneration of the vegetation in the following years after the calamity.

The results of the development of the individual monitored indices are documented in Figure 5. The evolution of the values of the monitored indices shows that the wind calamity that occurred in 2004 significantly affected the state of the forest vegetation. The examined indices strongly reflected this wind calamity. It can be seen in Figure 5 that the most significant change (decrease) is in the values of the following vegetation indices NDVI, NDMI and wNDII; the decline in the curve is less significant for the FMI, SR and TVI indices. These results can be also seen from the statistical information in the Figure 5b, similar figures were calculated for all sites. For areas with disturbance, there was detected a significant difference between minimum and maximum. In comparison, this statistical indicator was lower in the areas with minimal disturbance. These results were also achieved with standard deviation. Places with minimal disturbance had lower standard deviation. The area gradually began to be covered by the ascending deciduous woods (after harvesting the dead trees), especially *Sorbus aucuparia*. These young deciduous trees helped with the regeneration of the vegetation cover in the damaged area.

In 2005, there is missing value in Figure 5a. This is in case of clouds in the satellite image (Figures A1 and A2 in Appendix A). In this situation, linear interpolation was used, and the curve was connected to the nearest value point. The same method was used in other localities, if there was a problem with missing values due to cloud cover.

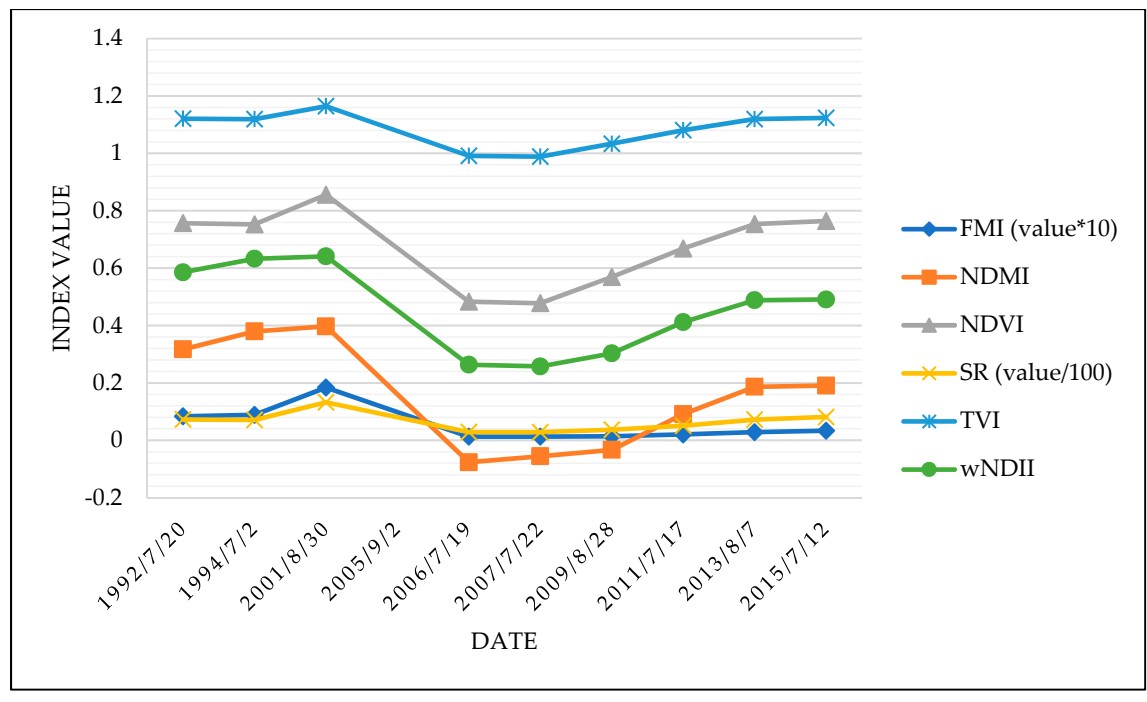

(a)

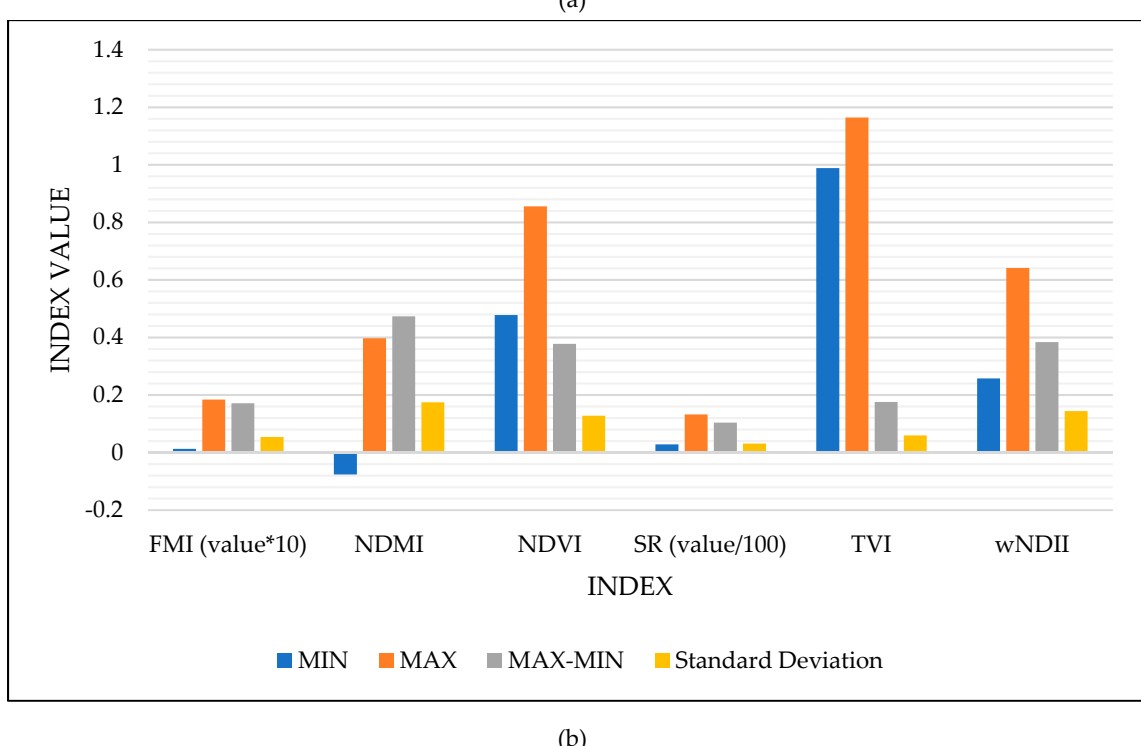

(b)

**Figure 5.** Time series of the vegetation indices for the areas affected by the wind calamity (Locality 1); (**a**) and the statistical information (**b**) (Source: Own work).

### 3.2. Locality 2

The second observed site is a site affected by the strongly expanding, devastating bark beetle after 2007. The forest vegetation after the attack was dying and left to spontaneously develop. The TS graph of the vegetation indices in this area can be seen in Figure 6.

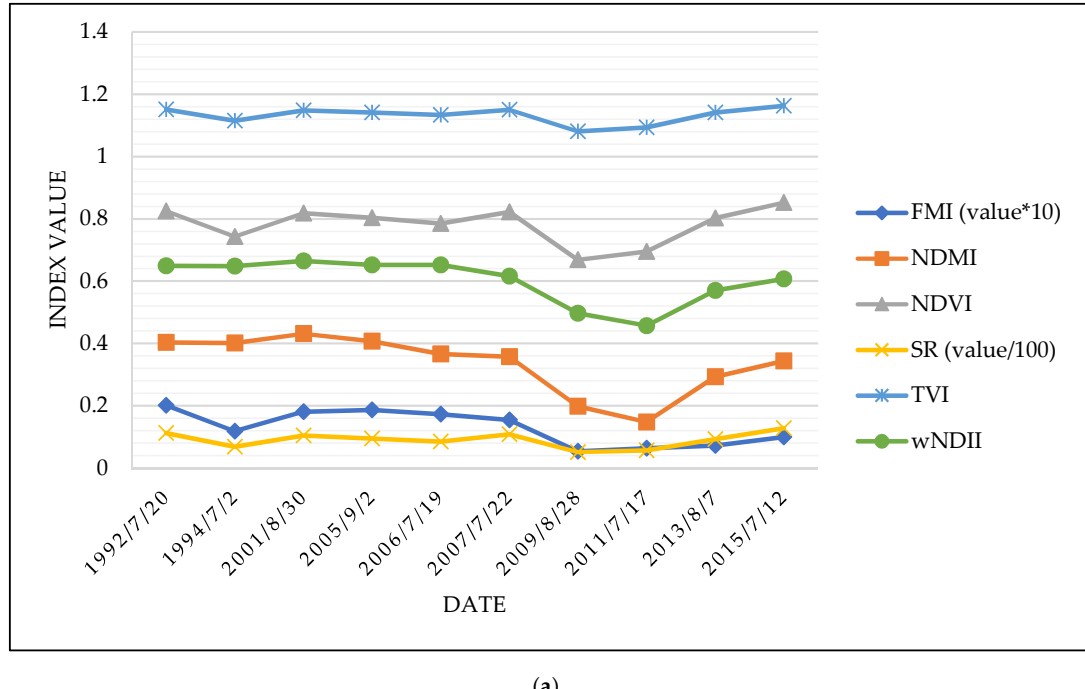

(**a**)

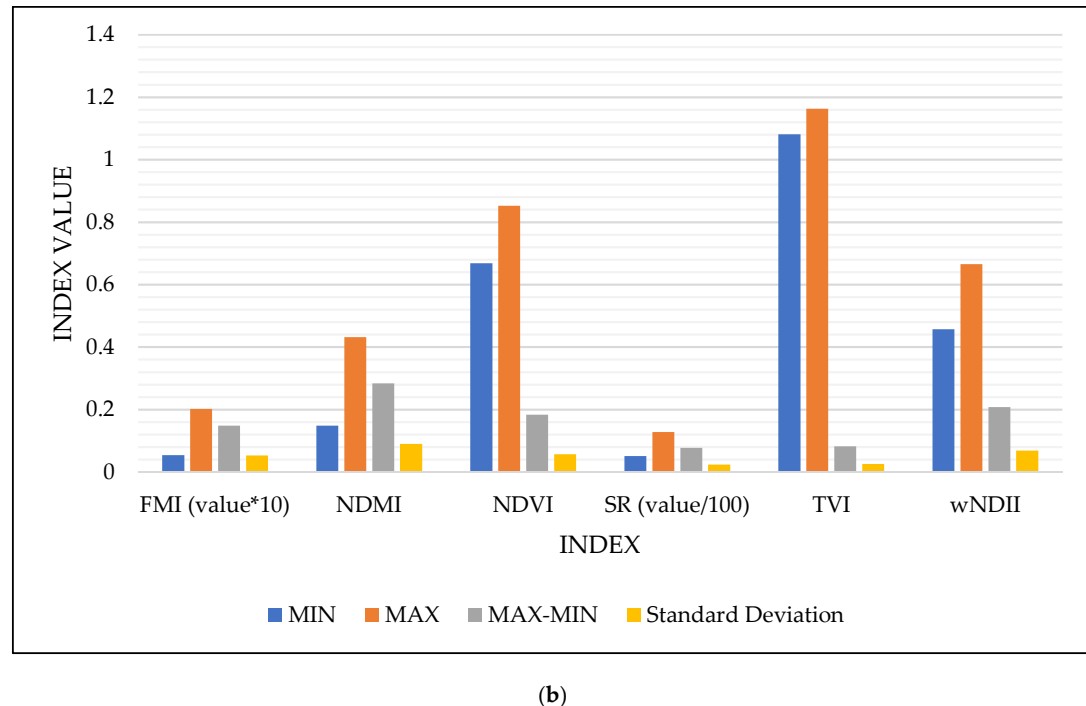

(**b**)

**Figure 6.** Time series of the vegetation indices for the area affected by the bark beetle (Locality 2) (**a**); and the statistical information (**b**) (Source: Own work).

In Figure 6, we can see some obvious trends. At the initiation phase of the bark beetle calamity, the vegetation indices wNDII, NDMI and FMI declined, while the values of the NDVI, SR and TVI indices slightly increased in the period 2005–2007. During 2007–2009, the location was characterized by the significant degradation of the spruce forest. All the monitored indices responded with a decrease in the values continuing until 2012. Then, the natural regeneration of the territory with the dominance of the deciduous trees began, which successively occupied the deforested areas in the territory of the calamity. This was mainly reflected by the NDMI, wNDII and NDVI indices.

### 3.3. Locality 3

The next observed locality in the Low Tatras represents a place where, unlike the two previous localities, there have been no significant changes in the vegetation during the observed period. The time series of the vegetation indices can be seen in Figure 7.

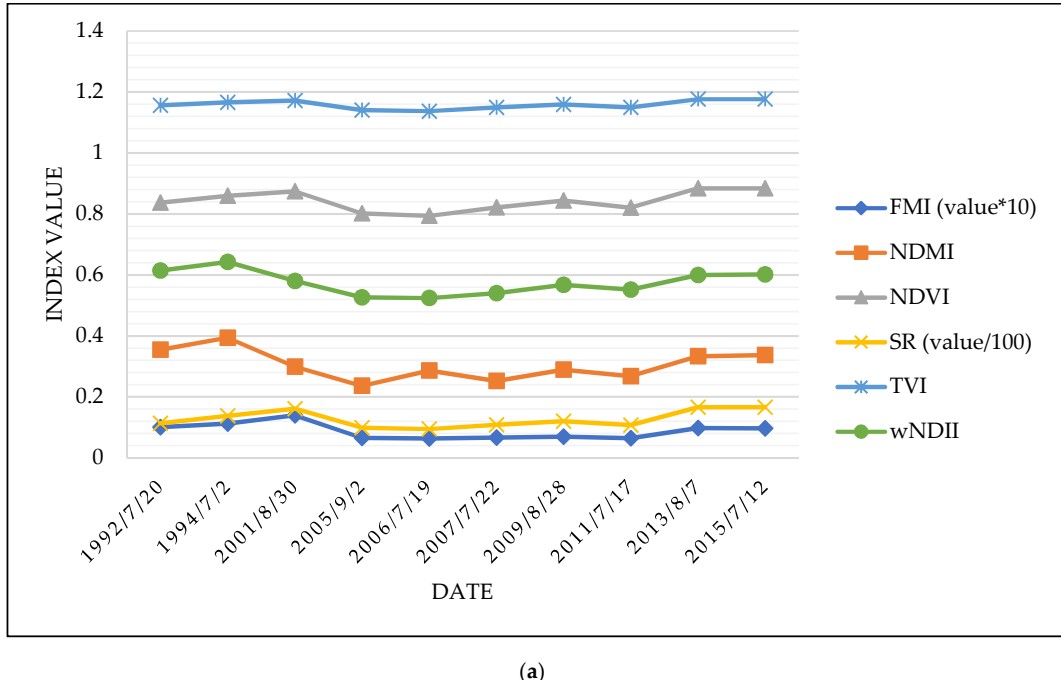

(**a**)

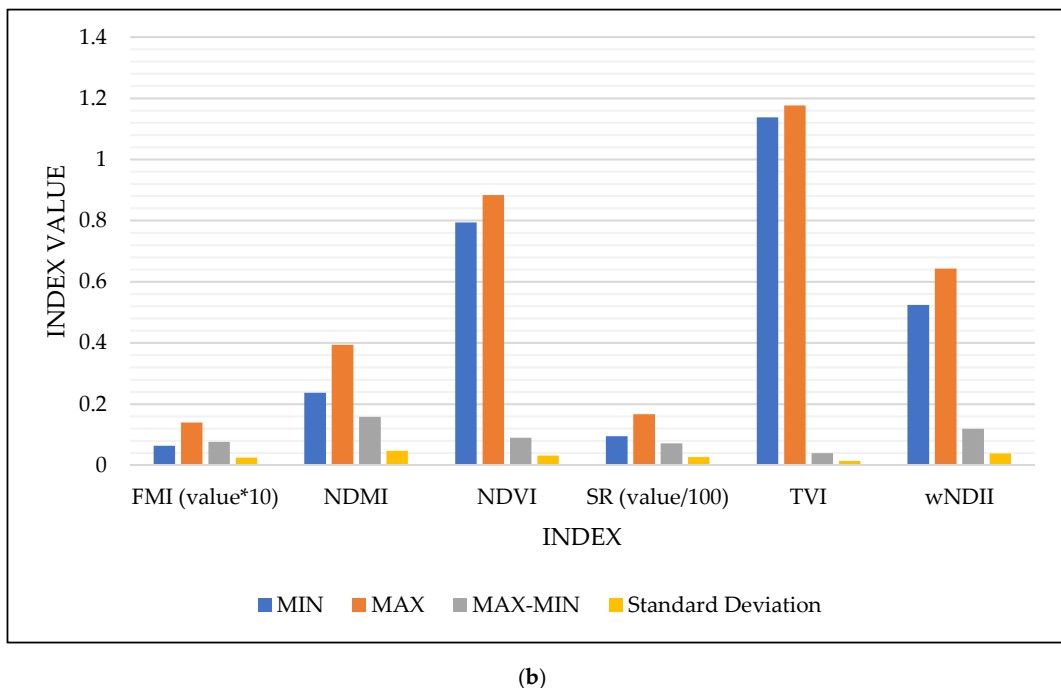

(**b**)

**Figure 7.** Time series of the vegetation indices for localities with minimal disturbance (**a**); and the statistical information (**b**) (Locality 3) (Source: Own work).

It can be seen in the figure that relatively small value changes occurred in the observed area. Insignificant oscillations are mainly evident in the wNDII and NDMI indices, as well as in the FMI. A slight fluctuation was recorded in the TVI index, less so in the NDVI and SR indices. In terms of

interpretation, it is necessary to realize that changes in the index values may reflect specific, local conditions, e.g., weather conditions: precipitation vs. drought [39].

### 3.4. Locality 4

The fourth observed site is a site affected by the strongly expanding, devastating bark beetle after 2001. The forest vegetation after the attack was dying and left to spontaneously develop. The TS graph of the vegetation indices in this area can be seen in Figure 8. All monitored indices responded with a decrease in the values with minimal values in 2005. Then, the natural regeneration of the territory was reflected by an increase of the values, mainly in NDMI, wNDII and NDVI indices.

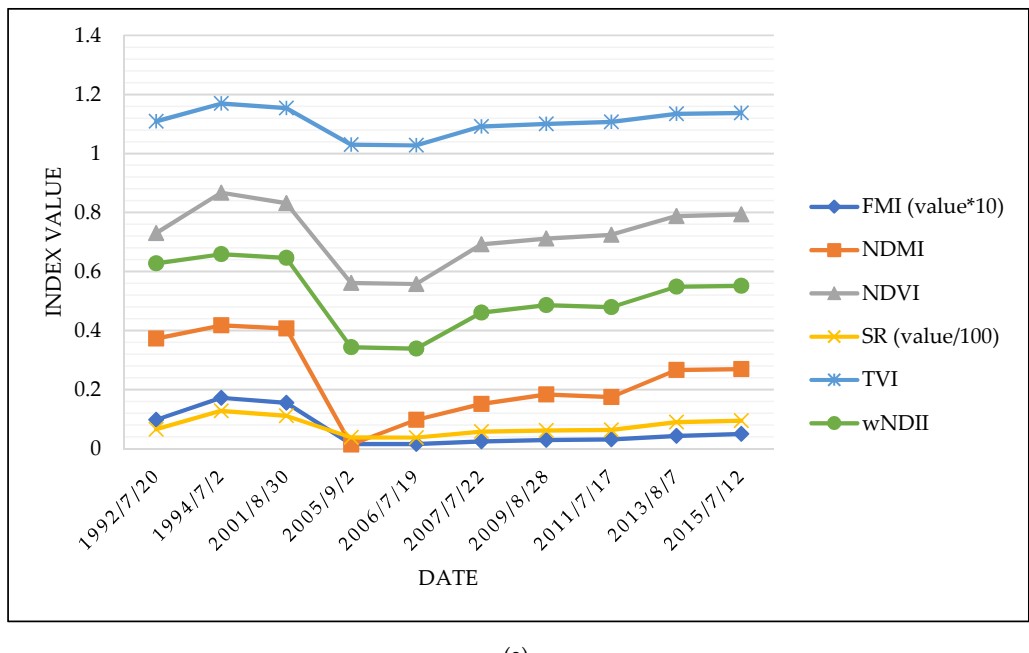

(**a**)

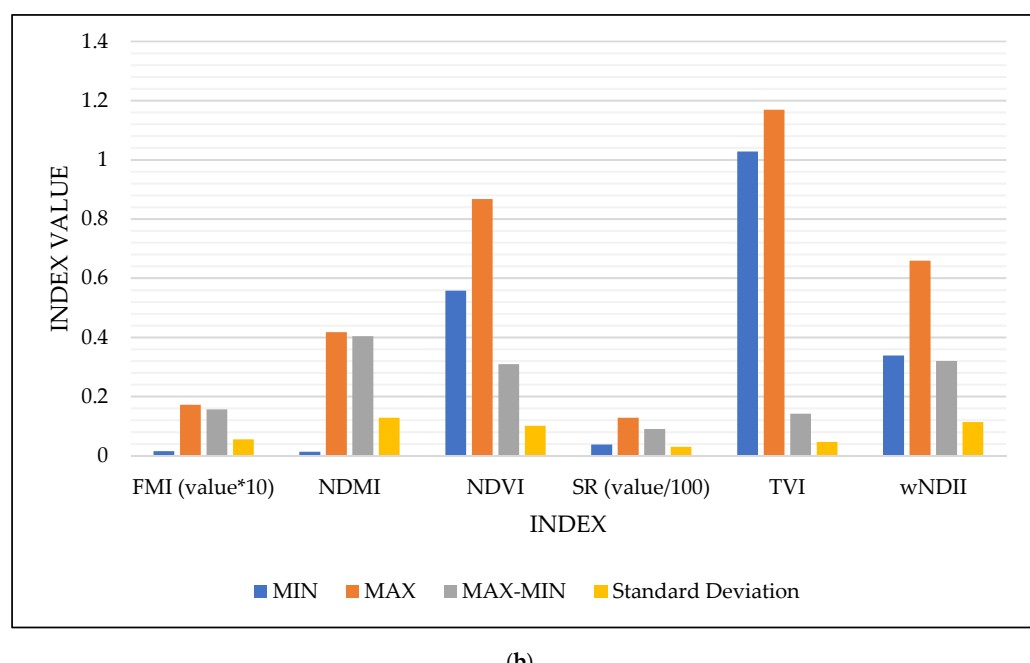

(**b**)

**Figure 8.** Time series of the vegetation indices for the area affected by the bark beetle and wind (**a**); and the statistical information (**b**) (Locality 4) (Source: Own work).

### 3.5. Locality 5

Locality 5 (Figure 9) in the Low Tatras represents a place where, unlike Localities 1, 2 and 4, there have been no significant changes in vegetation during the observed period. As can be seen in the graphs, values have a fixed level. Only in 2007 we can see an insignificant decline of values of the indices.

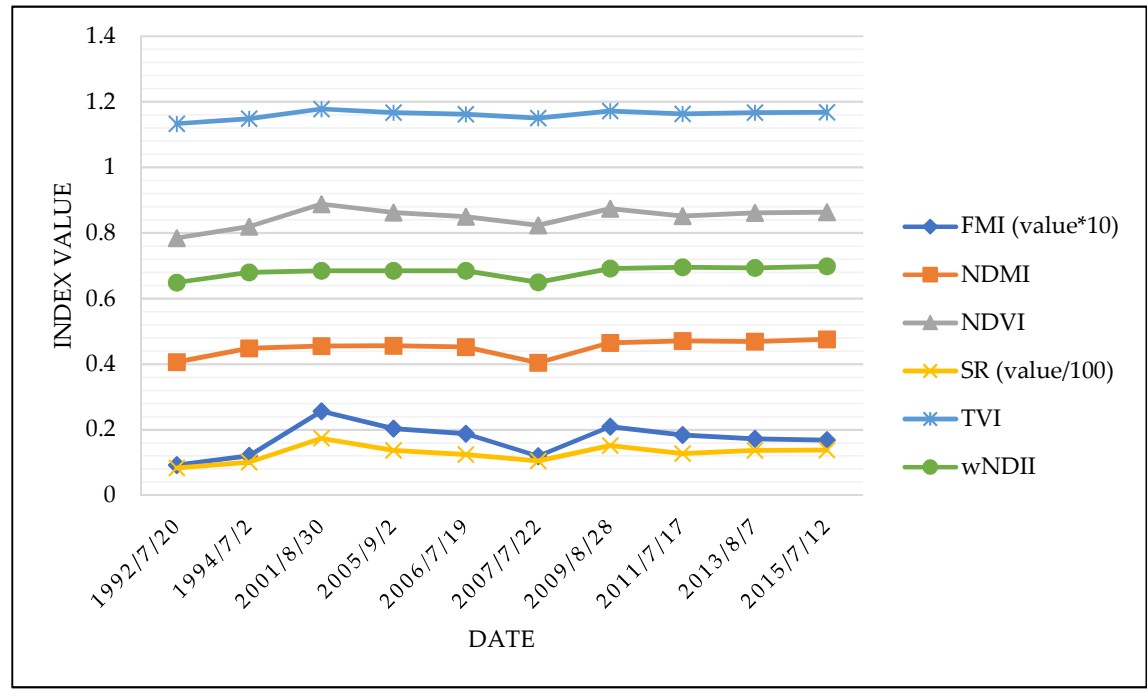

(**a**)

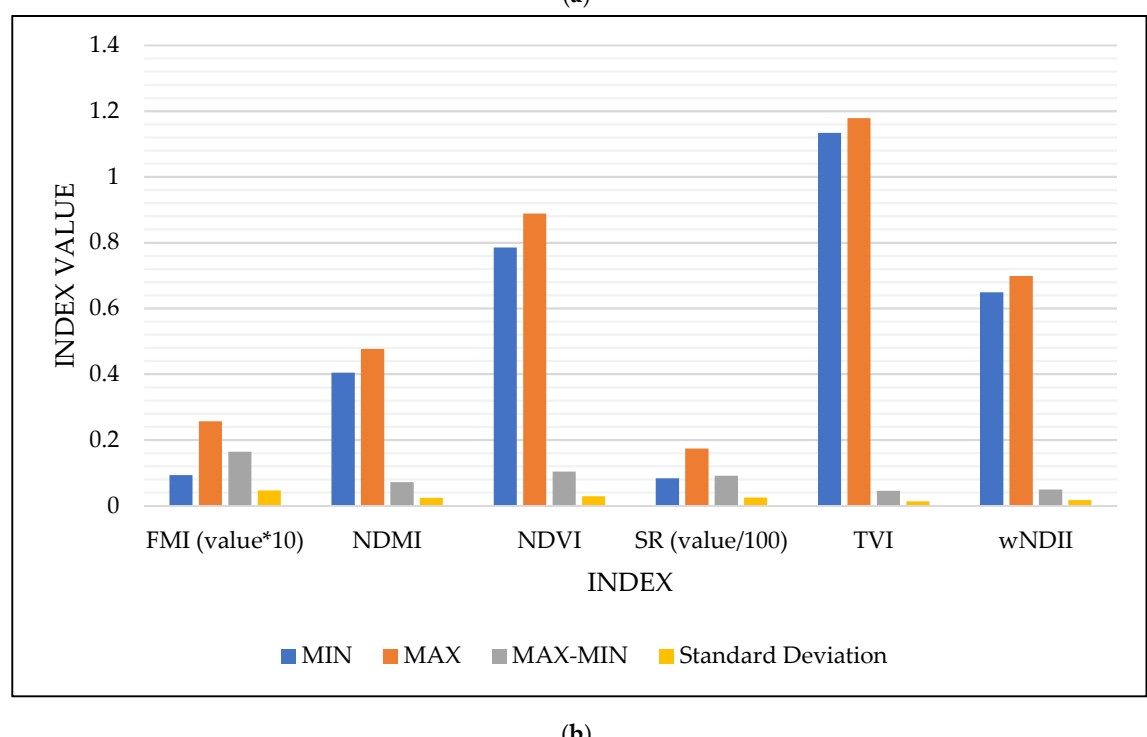

(**b**)

**Figure 9.** Time series of the locality with minimal disturbance (**a**); and the statistical information (**b**) (Locality 5) (Source: Own work).

### 3.6. Locality 6

The first site from the Sumava NP is a site that has been hit by a wind calamity (2007). After disturbing the tree structures, the forest was much more affected by bark beetle attack. The result can be seen in Figure 10. Here, the steep decline in the curves after 2008 is especially visible, caused by the disturbance of the trees by the wind and the subsequent bark beetle. The destruction process is so much faster than, for example, in Locality 7.

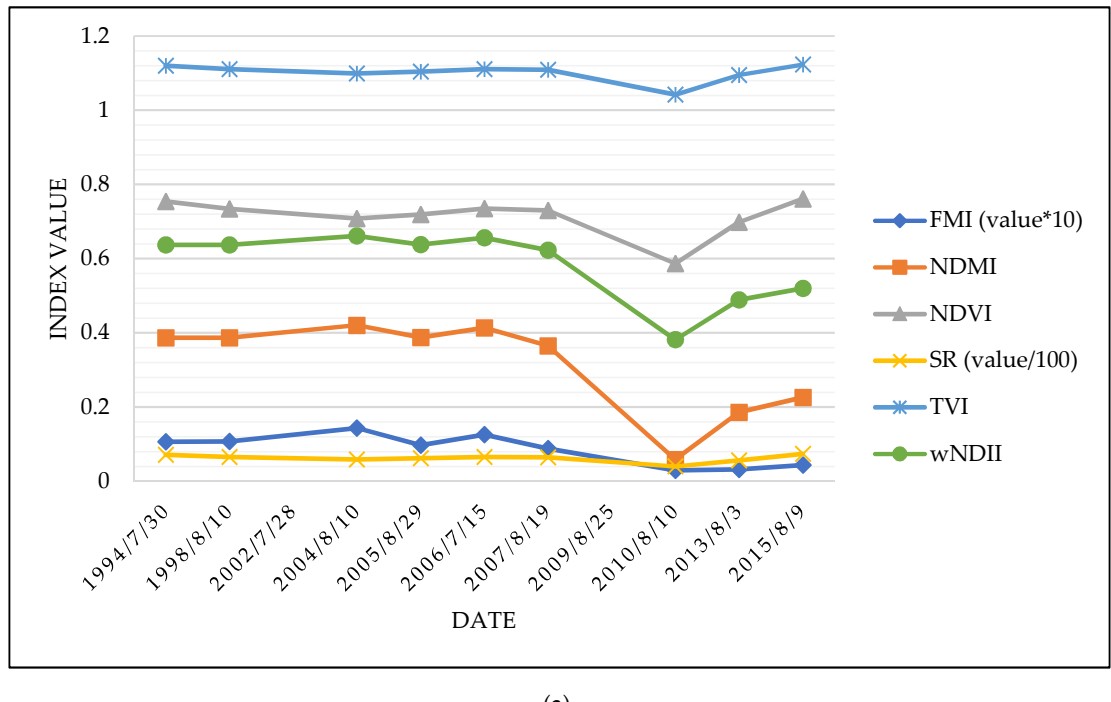

(**a**)

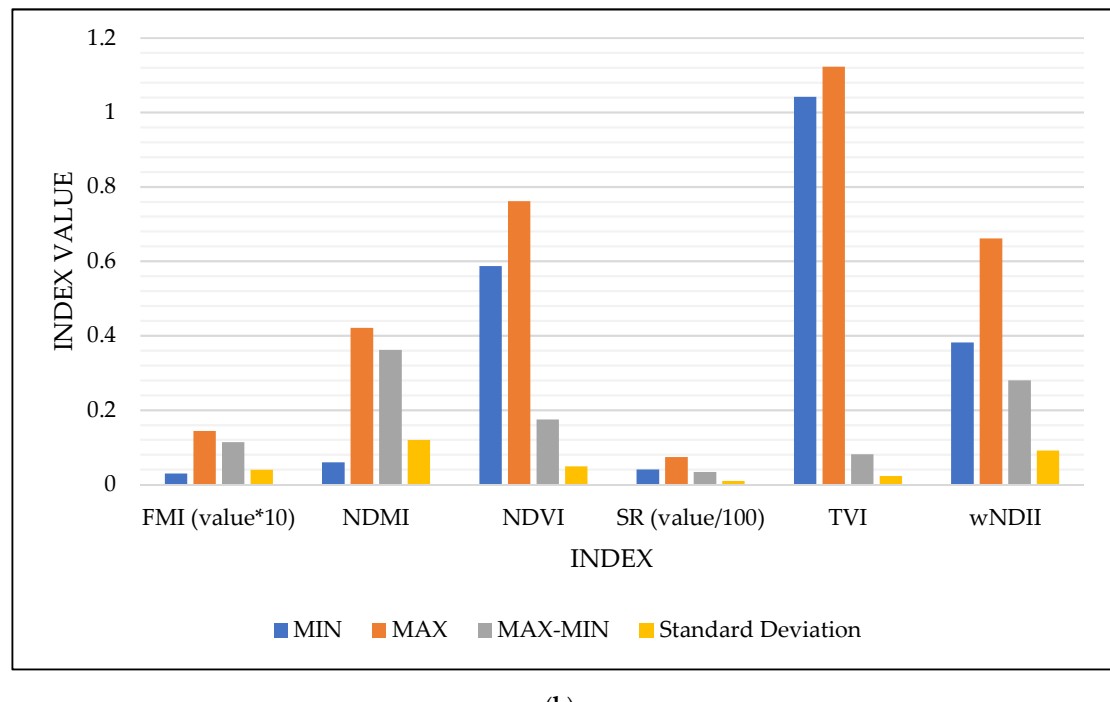

(**b**)

**Figure 10.** Time series of the vegetation indices for the area affected by the bark beetle and wind (**a**); and the statistical information (**b**) (Locality 6) (Source: Own work).

### 3.7. Locality 7

The next location from the Sumava NP is similar to Locality 2 from the Low Tatras region. This is a place where a biotic disturbance has occurred, i.e., a bark beetle calamity. Figure 11 shows the results of the time series analyses. The NDMI and wNDII indices reflected the whole disturbance event with a decreasing value. The NDVI, SR, and TVI indices had similar trends in the first years of the calamity (decrease). The dead trees were not harvested.

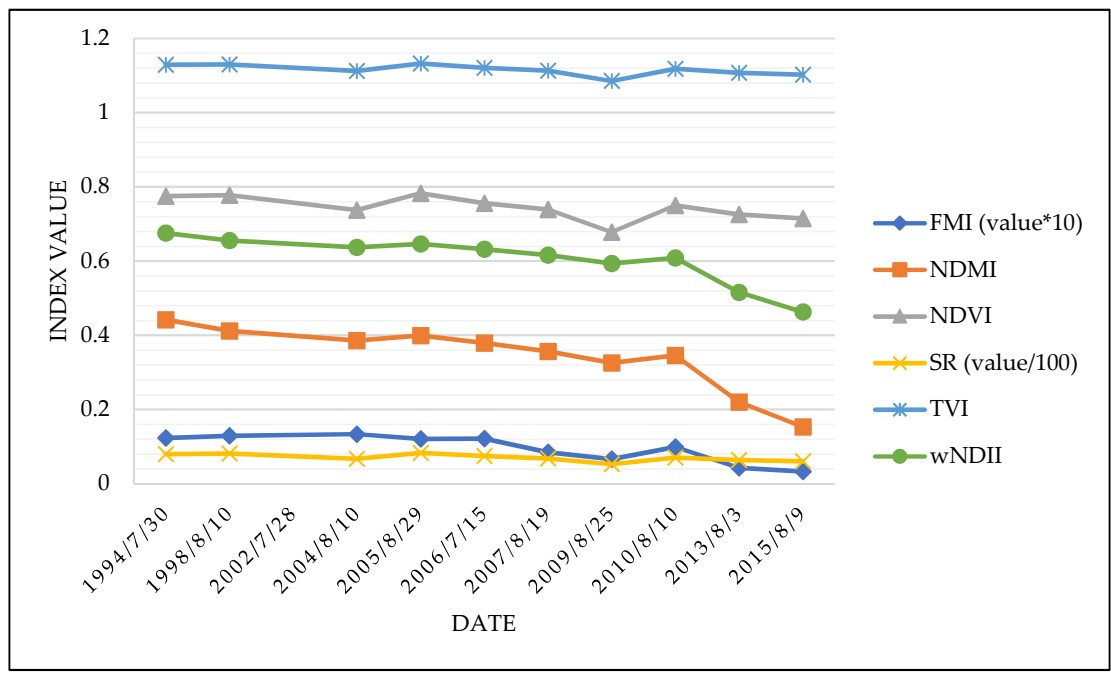

(**a**)

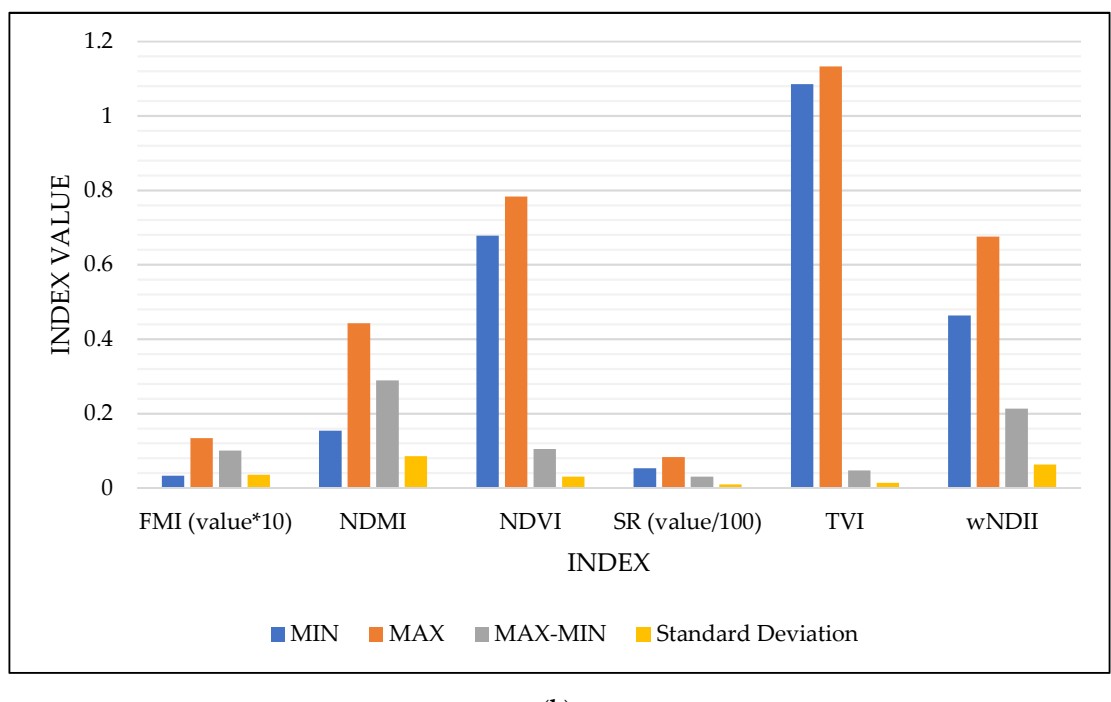

(**b**)

**Figure 11.** Time series of the vegetation indices for the area affected by the bark beetle (**a**); and the statistical information (**b**) (Locality 7) (Source: Own work).

### 3.8. Locality 8

The next site of interest is similar to Locality 3 in the Low Tatras NP: without any significant disturbance. This location is covered by a spruce forest with an approximate age of 50 years according to the in-situ data. In Figure 12, we can see the development with the oscillation in the values without any significant breaks.

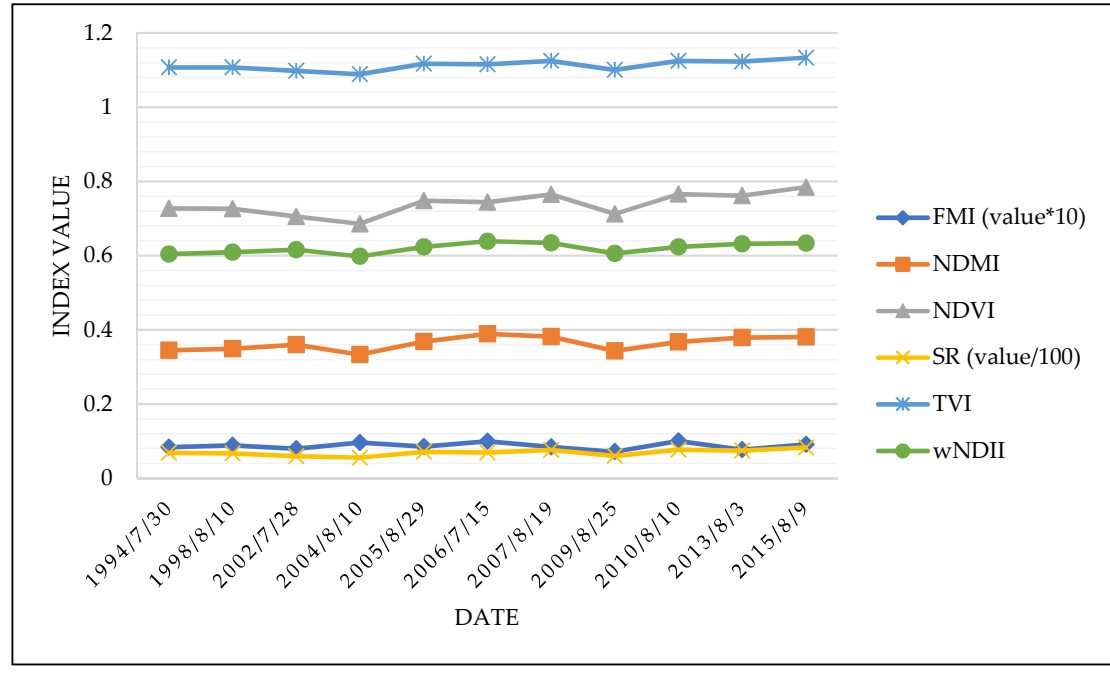

(**a**)

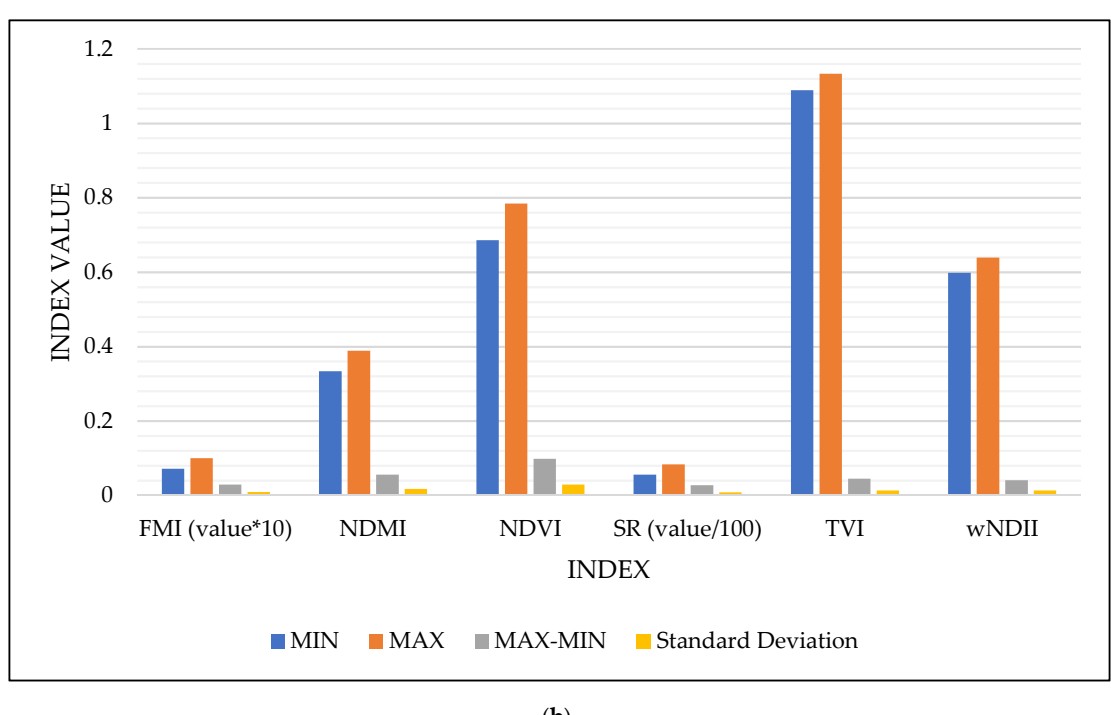

(**b**)

**Figure 12.** Time series of the locality with minimal disturbance (**a**); and the statistical information (**b**) (Locality 8) (Source: Own work).

### 3.9. Locality 9

The next location from the Sumava NP is similar to Locality 2 from the Low Tatras region and Locality 7 from the Sumava National Park. This is the place where a biotic disturbance has occurred, i.e., bark beetle calamity. Figure 13 shows the results of the time series analyses. The NDVI, NDMI and wNDII indices reflected the whole disturbance event with a decreasing value. The dead trees were not harvested after the disturbance.

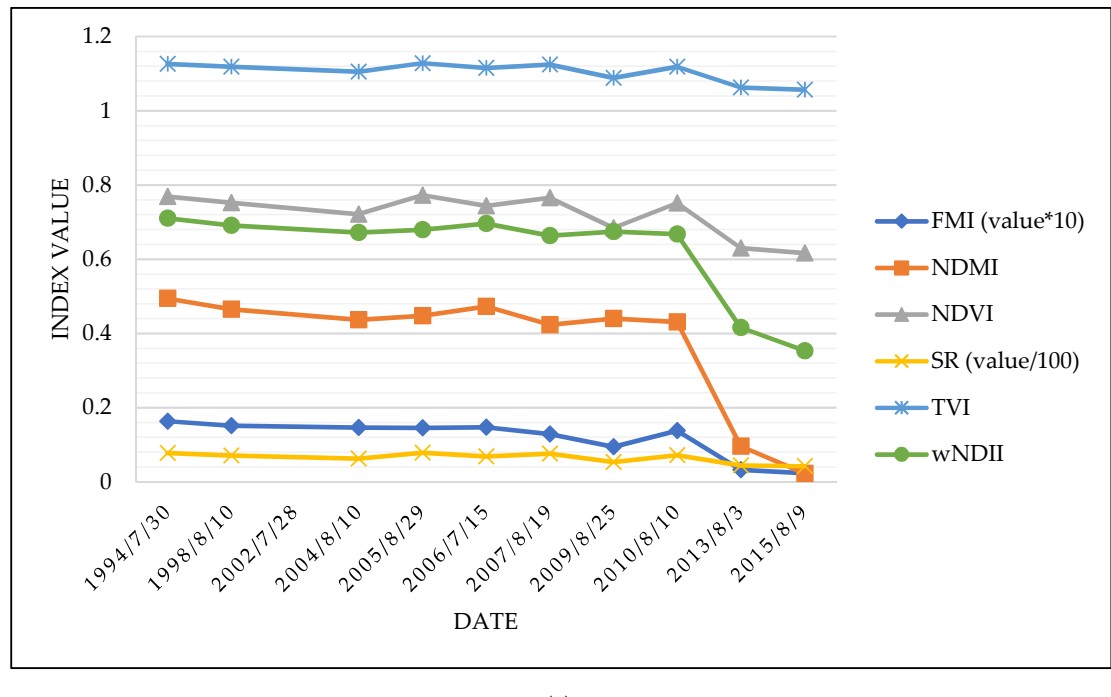

(**a**)

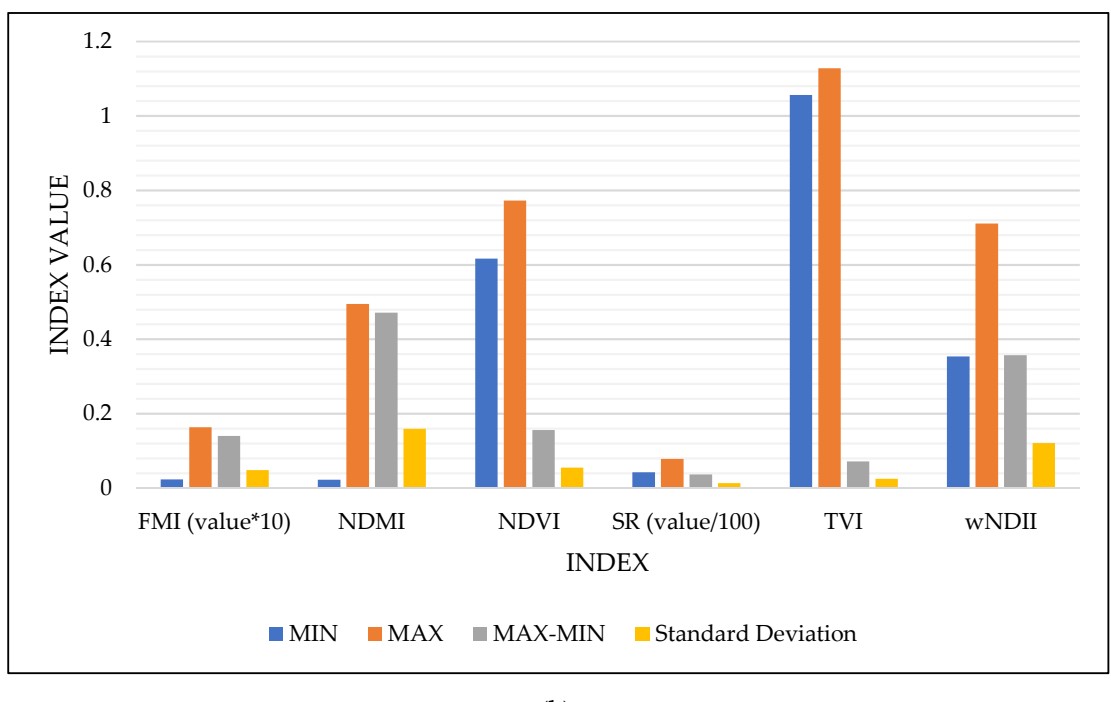

(**b**)

**Figure 13.** Time series of the vegetation indices for the area affected by the bark beetle (**a**); and the statistical information (**b**) (Locality 9) (Source: Own work).

### 3.10. Locality 10

The last site of interest is similar to Localities 3 and 5 in the Low Tatras NP (without any significant disturbance) and Locality 8 in the Sumava National Park (with minimal disturbance). This location is covered by a spruce forest with an approximate age of 60 years according to the in-situ data. In Figure 14, we can see the development with the oscillation in the values with only small breaks.

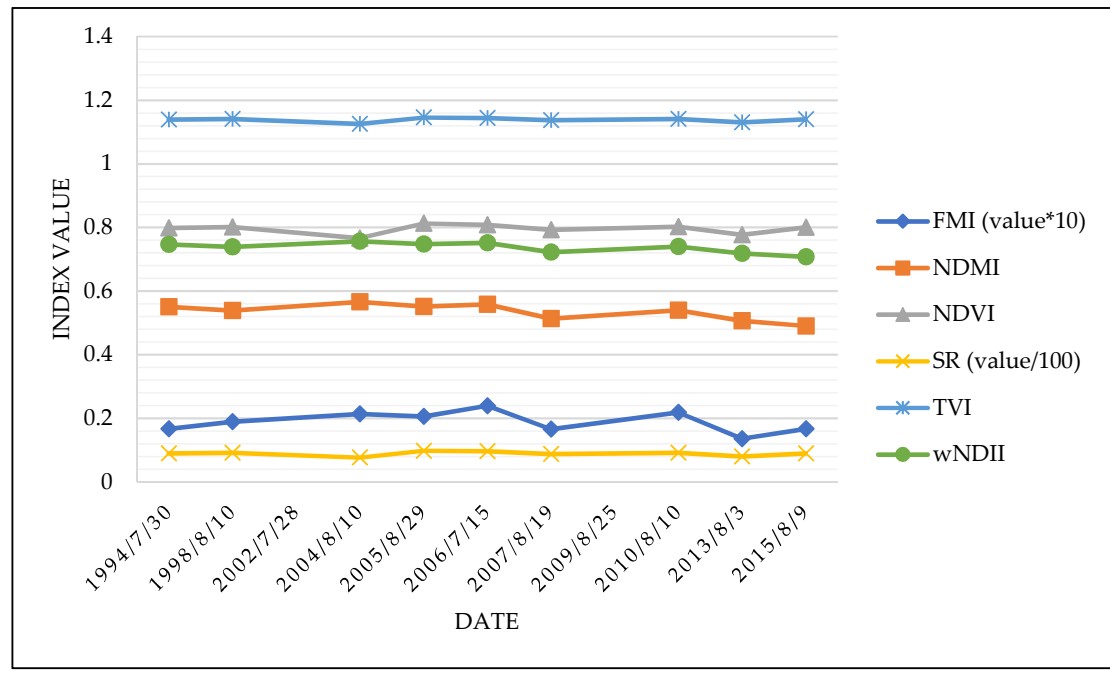

(**a**)

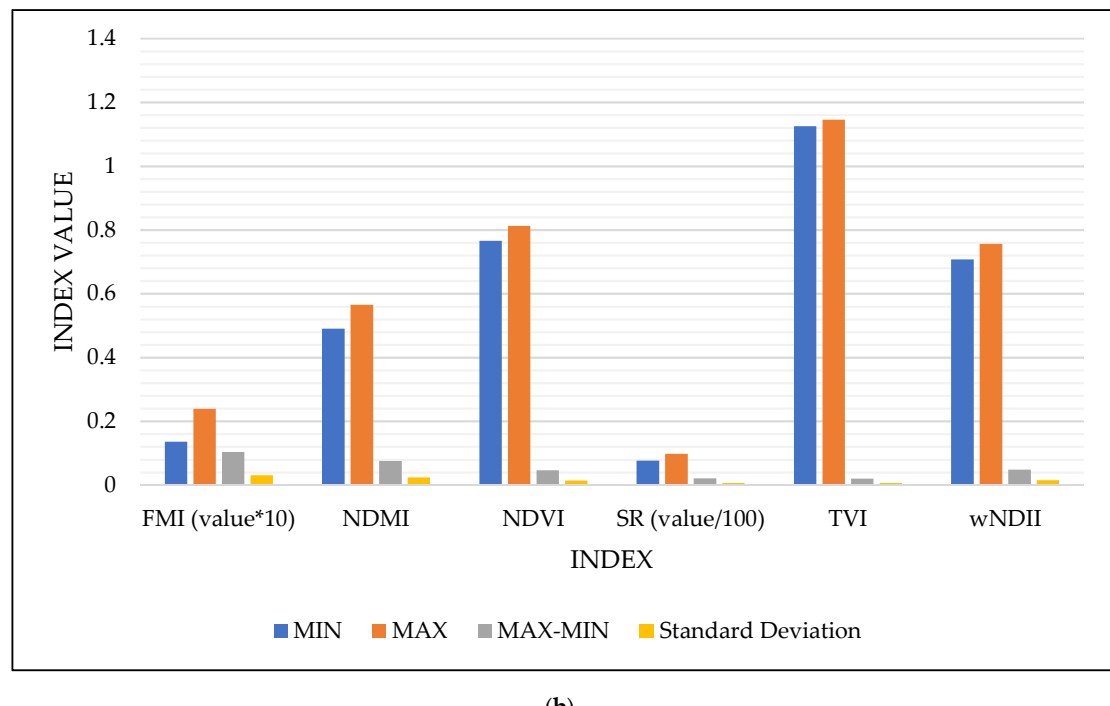

(**b**)

**Figure 14.** Time series of locality with minimal disturbance (**a**); and the statistical information (**b**) (Locality 10) (Source: Own work).

*3.11. Statistical Results and Comparison of Both Specific Areas*

Figures 15, 16, 17 and 18 show the average values from the Low Tatras National Park and the Sumava National Park. The graphs 16 and 18 show the minimal variance caused by the different site conditions. However, in Figures 15 and 17, we can see the graphs of both sites affected by the calamities with average values. For Figure 17 (the Sumava National Park), it can be stated that, due to the gradual spread of the disturbance in the individual years, the longer span of the time of the calamity breakdown shows the calamity break is slower. In addition to the summary graphs, the individual aggregate values of the individual vegetation indices are described in Tables 6 and 7 for the Low Tatras, and Tables 8 and 9 for the Sumava National Park.

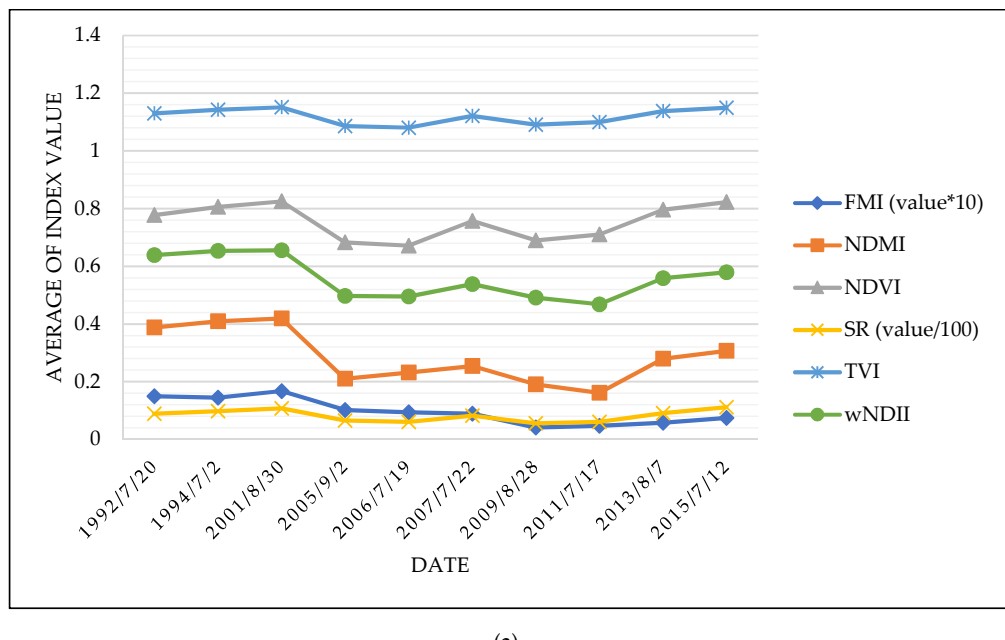

(**a**)

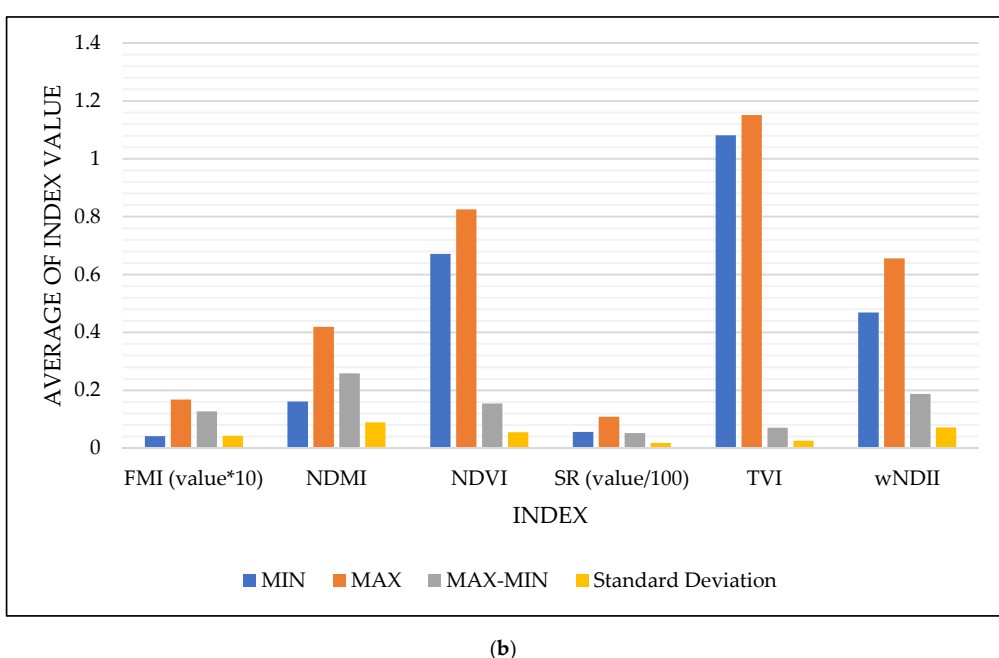

(**b**)

**Figure 15.** Time series of the vegetation indices for the area affected by the bark beetle and the wind calamity (**a**); and the statistical information (**b**) (the average from the Low Tatras National Park) (Source: Own work).

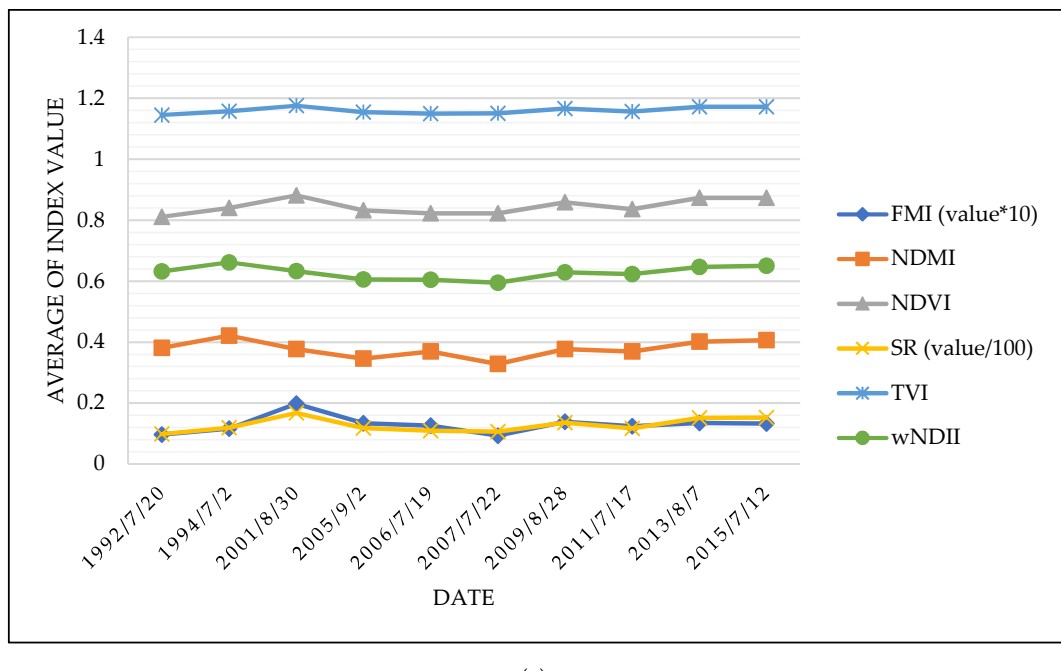

(**a**)

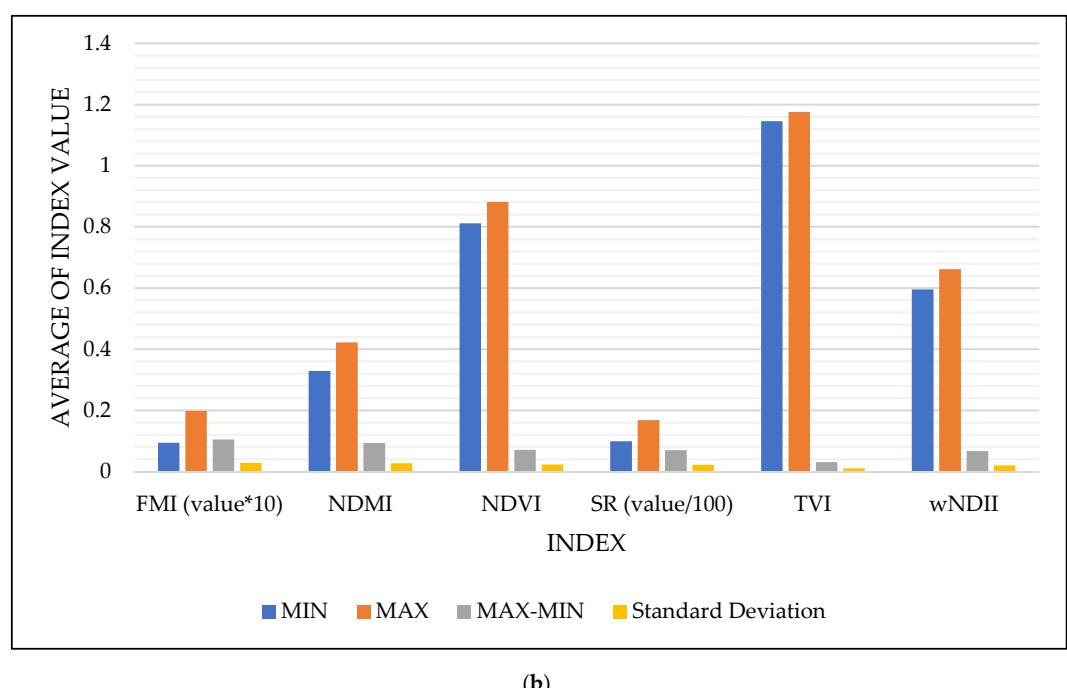

(**b**)

**Figure 16.** Time series of the vegetation indices of the areas with minimal disturbance (**a**); and the statistical information (**b**) (the average from the Low Tatras National Park) (Source: Own work).

**Table 6.** Statistical information about the average values in the disturbed areas in the Low Tatras NP.

| Vegetation Index | MIN | MAX | MAX-MIN | Standard Deviation |
|---|---|---|---|---|
| FMI (value * 10) | 0.0412 | 0.1676 | 0.1264 | 0.0421 |
| NDMI | 0.1612 | 0.4191 | 0.2579 | 0.0883 |
| NDVI | 0.6712 | 0.8249 | 0.1537 | 0.0551 |
| SR (value/100) | 0.0560 | 0.1078 | 0.0518 | 0.0177 |
| TVI | 1.0805 | 1.1510 | 0.0705 | 0.0252 |
| wNDII | 0.4683 | 0.6555 | 0.1872 | 0.0710 |

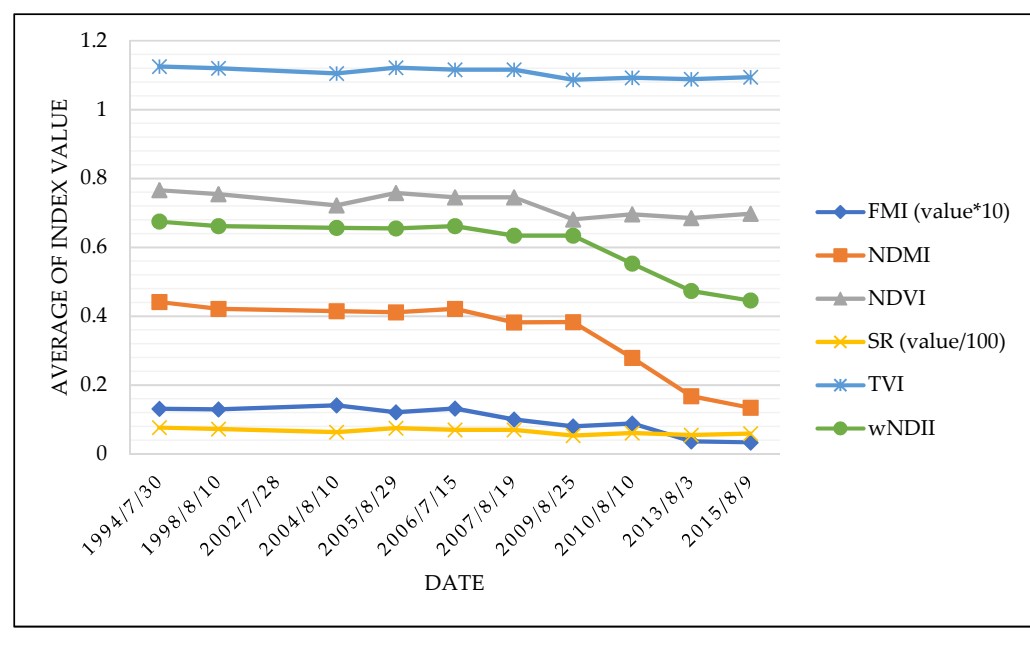

(**a**)

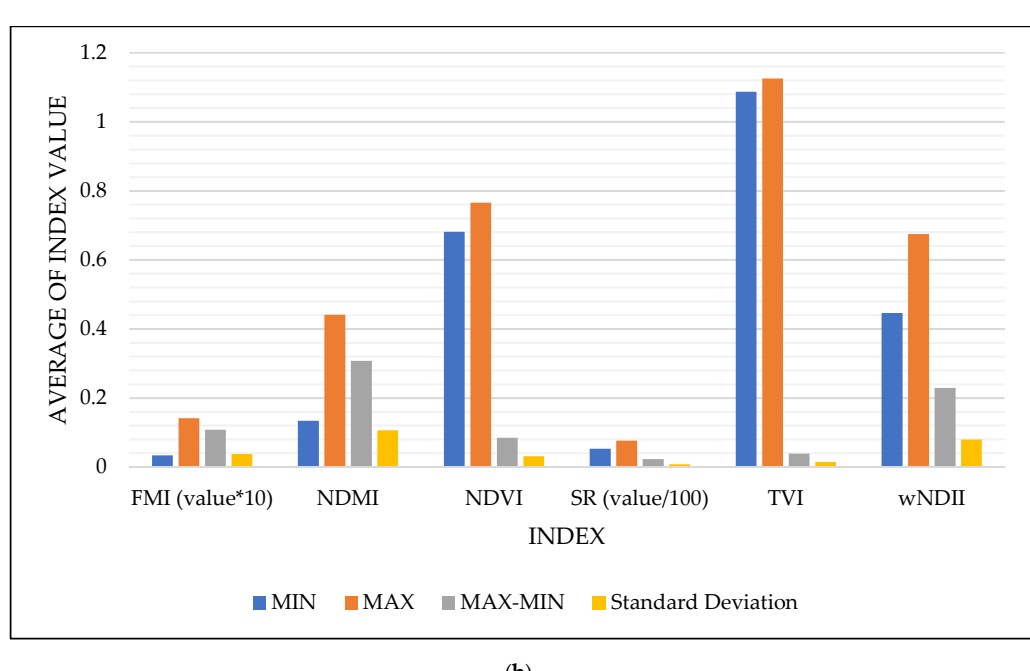

(**b**)

**Figure 17.** Time series of the vegetation indices for the area affected by the bark beetle and the wind calamity (**a**); and the statistical information (**b**) (the average from the Sumava National Park) (Source: Own work).

**Table 7.** Statistical information about the average values in the areas with minimal disturbance in the Low Tatras NP.

| Vegetation Index | MIN | MAX | MAX-MIN | Standard Deviation |
|---|---|---|---|---|
| FMI (value * 10) | 0.0932 | 0.1980 | 0.1048 | 0.0273 |
| NDMI | 0.3283 | 0.4213 | 0.0930 | 0.0262 |
| NDVI | 0.8108 | 0.8812 | 0.0705 | 0.0227 |
| SR (value/100) | 0.0984 | 0.1678 | 0.0694 | 0.0212 |
| TVI | 1.1448 | 1.1752 | 0.0304 | 0.0098 |
| wNDII | 0.5948 | 0.6614 | 0.0666 | 0.0200 |

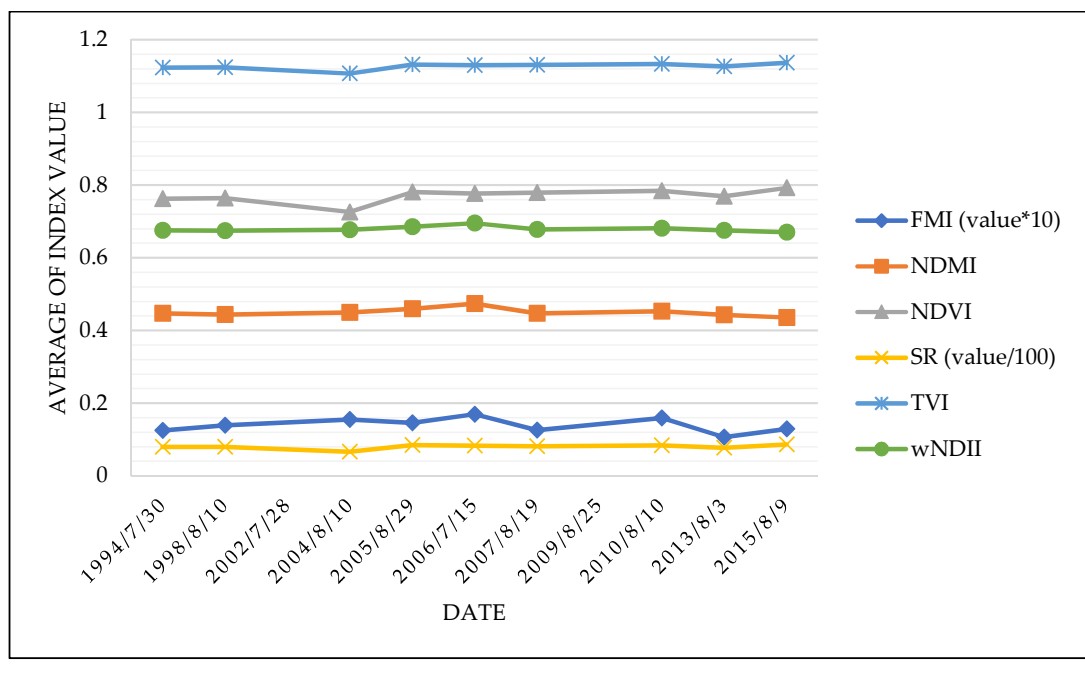

(**a**)

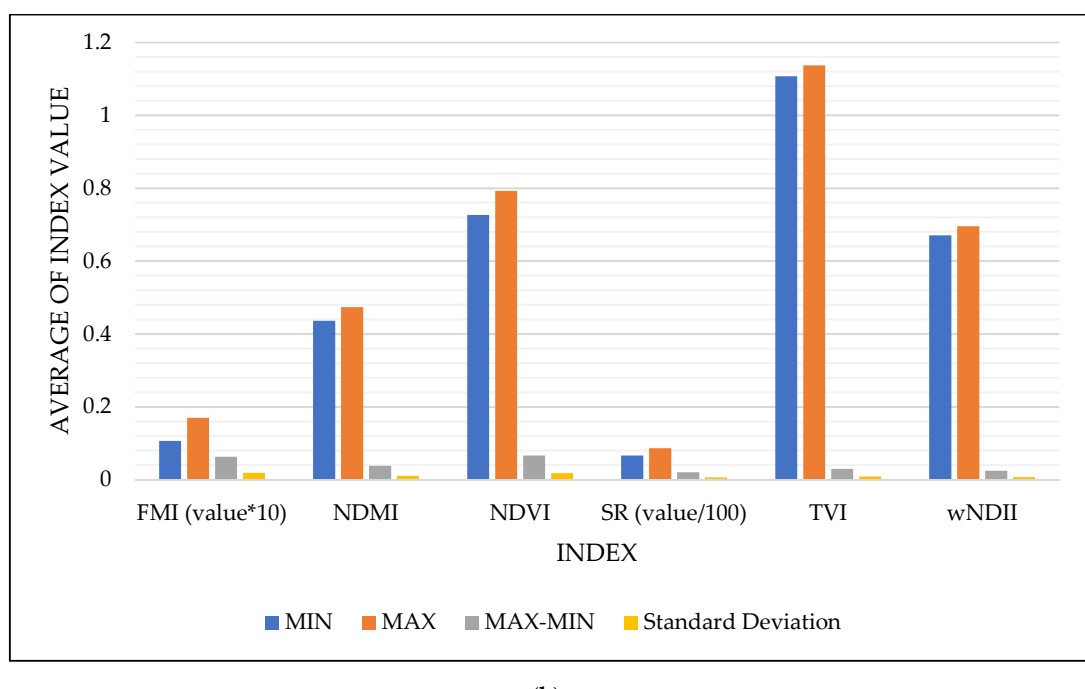

(**b**)

**Figure 18.** Time series of the vegetation indices of the areas with minimal disturbance (**a**); and the statistical information (**b**) (the average from the Sumava National Park) (Source: Own work).

**Table 8.** Statistical information about the average values in the disturbed areas in the Sumava NP.

| Vegetation Index | MIN | MAX | MAX-MIN | Standard Deviation |
|---|---|---|---|---|
| FMI (value * 10) | 0.0334 | 0.1411 | 0.1077 | 0.0374 |
| NDMI | 0.1341 | 0.4411 | 0.3070 | 0.1065 |
| NDVI | 0.6812 | 0.7658 | 0.0846 | 0.0309 |
| SR (value/100) | 0.0532 | 0.0762 | 0.0231 | 0.0079 |
| TVI | 1.0868 | 1.1250 | 0.0383 | 0.0141 |
| wNDII | 0.4454 | 0.6745 | 0.2291 | 0.0797 |

**Table 9.** Statistical information about the average values in the areas with minimal disturbance in the Sumava NP.

| Vegetation Index | MIN | MAX | MAX-MIN | Standard Deviation |
|---|---|---|---|---|
| FMI (value * 10) | 0.0716 | 0.1694 | 0.0978 | 0.0300 |
| NDMI | 0.3433 | 0.4737 | 0.1304 | 0.0395 |
| NDVI | 0.7051 | 0.7922 | 0.0871 | 0.0290 |
| SR (value/100) | 0.0593 | 0.0866 | 0.0273 | 0.0094 |
| TVI | 1.0976 | 1.1367 | 0.0391 | 0.0130 |
| wNDII | 0.6061 | 0.6952 | 0.0891 | 0.0271 |

## 4. Discussion and Conclusions

The forests in the Low Tatras and Sumava National Parks have been severely affected by wind calamities and, subsequently, by bark beetle insects. For the selected locations, which differed in the type of disturbance, the calculation of the vegetation indices and the comparison/evaluation of their trajectories were performed. The main aim was to determine the suitability of the individual vegetation indices for the detection of the different types of biotic and abiotic disturbances using a TS analysis. Based on the results of this study, it is obvious that each type of disturbance, whether biotic or abiotic, has a specific development and different consequences on the state (health) of the vegetation/spectral expression. In the case of a wind disaster, a large part of the forest vegetation is devastated in a quick period. This corresponds to a large impact decrease (more than with the bark beetle calamity, because the biotic calamity is decreasing more slowly) in the indices using RED, NIR and SWIR bands [45]. A forestry intervention or succession of new vegetation causes an increase in the reflection in the green and especially in the near infrared part of the spectrum in a relatively short time after the disturbance. A wind calamity has a strong effect on the vegetation, however the recovery in the following years is quick, both in a spontaneous way and a forest-controlled way. When the three sites affected by wind calamities in Low Tatras and Sumava NPs were compared, similar result could be seen. The vegetation indices based on the use of the SWIR band (e.g., the NDMI and wNDII index) show similar trends as well [46]. A bark beetle calamity has a much more complicated progress with a specific effect on the vegetation. Several generations of beetles gradually occupy the forest with a strong culmination phase. There can be three generations of beetles in one year, which greatly disturb the resistance of the forest stand. Our results confirmed the high sensitivity of the wNDII and NDMI vegetation indices based on the SWIR band in the case of the bark beetle disturbance (see Figure 17). These vegetation indices were able to reflect all the phases of the disturbance using the time series analysis. The reflectivity of the healthy forest vegetation in the SWIR was lower than in the NIR. However, during the disturbance/damage, the SWIR reflectivity tended to increase. It is clear from the results that this aspect played an important role in detecting the disturbed period (compare the NDMI or wNDII vegetation indices that use the SWIR band with the vegetation indices based on NIR such as NDVI). An important result of this study is that the NDMI and wNDII vegetation indices can detect all the stages of the bark calamity (before, during and after the culmination) and are thus suitable indicators in monitoring the health status of the forest vegetation in such specific/environmentally complex processes as a bark beetle calamity with its causes and consequences. On the other hand, it can be stated that indices using SWIR are more sensitive to more local factors [45]. The values of the vegetation indices can be affected by many factors out of our current investigation, e.g., meteorological factors such as precipitation or soil moisture. For a more detailed interpretation of these changes, it would be necessary to include several factors, such as meteorological, which was beyond the scope of this study.

In terms of comparison of the two observed areas (Sumava NP and Low Tatras NP), statistical summaries were made. It can be said that the calamity conditions of the observed sites took place over a longer period in Sumava areas. This can also be seen from the individual figures. However, if we look at aggregate indicators, both locations have a very different evolution of recovery phase. In particular,

in the Low Tatras, trees were harvested after the calamities. This results in a different development from the Sumava areas, where the trees were left. The spread of new trees and bushes in Sumava is slower and the values of the selected indices after overcoming the recovery phase do not reach their initial values from before the calamity, but in the Low Tatras in several cases indices approximately reach their initial values from before the calamity. However, this does not indicate anything about an unnatural interference with nature. In Figures 15–18 and Tables 6–9, the NDMI index can be picked up again as the best index to detect the disturbance of the forest ecosystems. Secondly, the wNDII index also had comparable results. The NDVI index also corresponded to the disturbances, but the problem of the NDVI index is that it cannot differentiate the recovery phase in comparison with the wNDII and NDMI indices.

A long-term TS can be used to track the development and current state of diverse forest ecosystems. The Landsat data are a very useful data source in this respect [47]. On the other hand, it is necessary to pay attention to the compatibility of the data coming from various types of Landsat sensors [40]. Although Landsat CDR data contain atmospheric and radiometric corrections, it is still useful to perform radiometric normalization. Thanks to the normalization, we can reduce the effects of the different times (phenology period) and places of acquisition, the different positions of the sun, and radiometric and spectral differences. The PIF Linear Based method appears to be a suitable standardization method. For data normalization of the CDR Landsat data, custom applications were developed in the MATLAB environment. The application allows one to work with any type of satellite data and it is transferable to a wide range of software and operating systems.

This study should serve as an example when the development of the impacts of the disturbances on the state of the forest vegetation was analyzed by traditional methodological procedures using vegetation indices. These results and methods should be useful, inspirational to the institutions for the management of the protected areas and to determine the appropriate forest management practices implemented in them. The time series results obtained can be applied to generate predictions of future states and to observe the spontaneous and economically managed forest regeneration. The use of satellite data provides progressive opportunities to monitor and evaluate the state and changes in the forest vegetation for the purpose of the protection and management of national parks [46,47].

**Author Contributions:** Conceptualization, P.S., R.H. and J.L.; Methodology, J.L. and P.S.; Software, J.L.; Validation, J.L., R.H., D.P. and P.S.; Formal Analysis, J.L. and R.H.; Investigation, J.L., R.H., D.P. and P.S.; Resources, J.L. and D.P.; Data Curation, J.L. and R.H.; Writing-Original Draft Preparation, J.L. and P.S.; Writing-Review & Editing, J.L., P.S. and R.H.; Visualization, J.L. and D.P.; Supervision, P.S.

**Funding:** This research was funded by the Grant Agency of the Charles University (GAUK, Charles Univ, Fac Sci) grant number 512217 (2017–2019): "Hodnoceni vlivu disturbanci na lesni ekosystemy v Cesku a na Slovensku pomoci metod DPZ" and UNCE "Charles University Research Centre program UNCE/HUM/018".

**Acknowledgments:** We would like to thank for the support of the Grant Agency of the Charles University (GAUK) and Charles University Research Centre program UNCE. The authors thank the anonymous reviewers for their critical review and constructive comments.

**Conflicts of Interest:** The authors declare no conflict of interest.

# Appendix A

*Statistical Results for the Localities in the Low Tatras National Park*

**Locality 1**

| Index/Date | 20.07.1992 | 02.07.1994 | 30.08.2001 | 02.09.2005 | 19.07.2006 | 22.07.2007 | 28.08.2009 | 17.07.2011 | 07.08.2013 | 12.07.2015 | MIN | MAX | MAX-MIN | St. Deviation |
|---|---|---|---|---|---|---|---|---|---|---|---|---|---|---|
| FMI (value*10) | 0,0836 | 0,0885 | 0,1838 | No Data (Clouds) | 0,0127 | 0,0127 | 0,0142 | 0,0205 | 0,0284 | 0,0333 | 0,0127 | 0,1838 | 0,1711 | 0,0539 |
| NDMI | 0,3169 | 0,3795 | 0,3971 | | -0,0765 | -0,0556 | -0,0334 | 0,0911 | 0,1865 | 0,1907 | -0,0765 | 0,3971 | 0,4736 | 0,1748 |
| NDVI | 0,7567 | 0,7525 | 0,8556 | | 0,4829 | 0,4777 | 0,569 | 0,6682 | 0,7535 | 0,7641 | 0,4777 | 0,8556 | 0,3779 | 0,1276 |
| SR (value/100) | 0,0726 | 0,0711 | 0,1324 | | 0,0287 | 0,0283 | 0,0369 | 0,0505 | 0,072 | 0,0813 | 0,0283 | 0,1324 | 0,1041 | 0,0309 |
| TVI | 1,1210 | 1,1191 | 1,1643 | | 0,9914 | 0,9887 | 1,0337 | 1,0808 | 1,1195 | 1,1237 | 0,9887 | 1,1643 | 0,1756 | 0,0597 |
| wNDII | 0,5857 | 0,6327 | 0,6412 | | 0,2635 | 0,2575 | 0,3031 | 0,4118 | 0,4882 | 0,4907 | 0,2575 | 0,6412 | 0,3837 | 0,1439 |

**Locality 2**

| Index/Date | 20.07.1992 | 02.07.1994 | 30.08.2001 | 02.09.2005 | 19.07.2006 | 22.07.2007 | 28.08.2009 | 17.07.2011 | 07.08.2013 | 12.07.2015 | MIN | MAX | MAX-MIN | St. Deviation |
|---|---|---|---|---|---|---|---|---|---|---|---|---|---|---|
| FMI (value*10) | 0,2014 | 0,1180 | 0,1803 | 0,1870 | 0,1724 | 0,1539 | 0,0536 | 0,0635 | 0,0720 | 0,0996 | 0,0536 | 0,2014 | 0,1478 | 0,0529 |
| NDMI | 0,4031 | 0,4012 | 0,4311 | 0,4072 | 0,3666 | 0,3574 | 0,1984 | 0,1479 | 0,2926 | 0,3437 | 0,1479 | 0,4311 | 0,2832 | 0,0897 |
| NDVI | 0,8251 | 0,7434 | 0,8179 | 0,8038 | 0,7848 | 0,8224 | 0,6683 | 0,6955 | 0,8030 | 0,8520 | 0,6683 | 0,8520 | 0,1837 | 0,0570 |
| SR (value/100) | 0,1119 | 0,0683 | 0,1047 | 0,0946 | 0,0850 | 0,1084 | 0,0511 | 0,0571 | 0,0926 | 0,1279 | 0,0511 | 0,1279 | 0,0768 | 0,0236 |
| TVI | 1,1510 | 1,1150 | 1,1479 | 1,1417 | 1,1334 | 1,1499 | 1,0807 | 1,0932 | 1,1414 | 1,1627 | 1,0807 | 1,1627 | 0,0820 | 0,0255 |
| wNDII | 0,6496 | 0,6478 | 0,6647 | 0,6518 | 0,6518 | 0,6165 | 0,4974 | 0,4572 | 0,5698 | 0,6072 | 0,4572 | 0,6647 | 0,2075 | 0,0681 |

**Locality 3**

| Index/Date | 20.07.1992 | 02.07.1994 | 30.08.2001 | 02.09.2005 | 19.07.2006 | 22.07.2007 | 28.08.2009 | 17.07.2011 | 07.08.2013 | 12.07.2015 | MIN | MAX | MAX-MIN | St. Deviation |
|---|---|---|---|---|---|---|---|---|---|---|---|---|---|---|
| FMI (value*10) | 0,1004 | 0,1124 | 0,1397 | 0,0656 | 0,0634 | 0,0665 | 0,0691 | 0,0650 | 0,0981 | 0,0968 | 0,0634 | 0,1397 | 0,0763 | 0,0246 |
| NDMI | 0,3548 | 0,3939 | 0,2987 | 0,2363 | 0,2862 | 0,2524 | 0,2894 | 0,2675 | 0,3335 | 0,3368 | 0,2363 | 0,3939 | 0,1576 | 0,0466 |
| NDVI | 0,8370 | 0,8595 | 0,8743 | 0,8015 | 0,7935 | 0,8217 | 0,8437 | 0,8203 | 0,8834 | 0,8835 | 0,7935 | 0,8835 | 0,0900 | 0,0311 |
| SR (value/100) | 0,1134 | 0,1378 | 0,1618 | 0,0992 | 0,0946 | 0,1091 | 0,1200 | 0,1079 | 0,1660 | 0,1662 | 0,0946 | 0,1662 | 0,0716 | 0,0267 |
| TVI | 1,1563 | 1,1659 | 1,1722 | 1,1405 | 1,1370 | 1,1495 | 1,1591 | 1,1489 | 1,1762 | 1,1762 | 1,1370 | 1,1762 | 0,0392 | 0,0135 |
| wNDII | 0,6146 | 0,6426 | 0,5803 | 0,5269 | 0,5241 | 0,5398 | 0,5677 | 0,5514 | 0,5998 | 0,6021 | 0,5241 | 0,6426 | 0,1185 | 0,0378 |

**Locality 4**

| Index/Date | 20.07.1992 | 02.07.1994 | 30.08.2001 | 02.09.2005 | 19.07.2006 | 22.07.2007 | 28.08.2009 | 17.07.2011 | 07.08.2013 | 12.07.2015 | MIN | MAX | MAX-MIN | St. Deviation |
|---|---|---|---|---|---|---|---|---|---|---|---|---|---|---|
| FMI (value*10) | 0,0971 | 0,1720 | 0,1549 | 0,0158 | 0,0157 | 0,0247 | 0,0288 | 0,0312 | 0,0430 | 0,0496 | 0,0157 | 0,1720 | 0,1563 | 0,0549 |
| NDMI | 0,3730 | 0,4174 | 0,4070 | 0,0134 | 0,0976 | 0,1515 | 0,1834 | 0,1745 | 0,2659 | 0,2690 | 0,0134 | 0,4174 | 0,4040 | 0,1286 |
| NDVI | 0,7297 | 0,8672 | 0,8318 | 0,5614 | 0,5576 | 0,6922 | 0,7114 | 0,7245 | 0,7881 | 0,7931 | 0,5576 | 0,8672 | 0,3096 | 0,1007 |
| SR (value/100) | 0,0654 | 0,1279 | 0,1109 | 0,0377 | 0,0373 | 0,0570 | 0,0609 | 0,0636 | 0,0899 | 0,0947 | 0,0373 | 0,1279 | 0,0906 | 0,0294 |
| TVI | 1,1088 | 1,1693 | 1,1540 | 1,0294 | 1,0275 | 1,0915 | 1,1004 | 1,1065 | 1,1347 | 1,1368 | 1,0275 | 1,1693 | 0,1418 | 0,0463 |
| wNDII | 0,6276 | 0,6585 | 0,6463 | 0,3434 | 0,3386 | 0,4607 | 0,4862 | 0,4794 | 0,5485 | 0,5510 | 0,3386 | 0,6585 | 0,3199 | 0,1140 |

**Locality 5**

| Index/Date | 20.07.1992 | 02.07.1994 | 30.08.2001 | 02.09.2005 | 19.07.2006 | 22.07.2007 | 28.08.2009 | 17.07.2011 | 07.08.2013 | 12.07.2015 | MIN | MAX | MAX-MIN | St. Deviation |
|---|---|---|---|---|---|---|---|---|---|---|---|---|---|---|
| FMI (value*10) | 0,0926 | 0,1204 | 0,2563 | 0,2031 | 0,1883 | 0,1199 | 0,2094 | 0,184 | 0,1722 | 0,1684 | 0,0926 | 0,2563 | 0,1637 | 0,0464 |
| NDMI | 0,4065 | 0,4486 | 0,4552 | 0,4561 | 0,4527 | 0,4041 | 0,4651 | 0,4713 | 0,4694 | 0,4759 | 0,4041 | 0,4759 | 0,0718 | 0,0241 |
| NDVI | 0,7845 | 0,8196 | 0,8881 | 0,8628 | 0,8501 | 0,8238 | 0,8739 | 0,8519 | 0,8617 | 0,8635 | 0,7845 | 0,8881 | 0,1036 | 0,0289 |
| SR (value/100) | 0,0834 | 0,1014 | 0,1738 | 0,1373 | 0,1245 | 0,1046 | 0,1517 | 0,1271 | 0,1372 | 0,1385 | 0,0834 | 0,1738 | 0,0904 | 0,0249 |
| TVI | 1,1333 | 1,1487 | 1,1782 | 1,1674 | 1,1619 | 1,1505 | 1,1721 | 1,1627 | 1,1669 | 1,1676 | 1,1333 | 1,1782 | 0,0449 | 0,0125 |
| wNDII | 0,6491 | 0,6802 | 0,6847 | 0,6851 | 0,6851 | 0,6498 | 0,6911 | 0,694 | 0,6982 | | 0,6491 | 0,6982 | 0,0491 | 0,0168 |

**Average: Disturbed Area**

| Index/Date | 20.07.1992 | 02.07.1994 | 30.08.2001 | 02.09.2005 | 19.07.2006 | 22.07.2007 | 28.08.2009 | 17.07.2011 | 07.08.2013 | 12.07.2015 | MIN | MAX | MAX-MIN | St. Deviation |
|---|---|---|---|---|---|---|---|---|---|---|---|---|---|---|
| FMI (value*10) | 0,1493 | 0,1450 | 0,1676 | 0,1014 | 0,0941 | 0,0893 | 0,0412 | 0,0474 | 0,0575 | 0,0746 | 0,0412 | 0,1676 | 0,1264 | 0,0421 |
| NDMI | 0,3881 | 0,4093 | 0,4191 | 0,2103 | 0,2321 | 0,2545 | 0,1909 | 0,1612 | 0,2793 | 0,3064 | 0,1612 | 0,4191 | 0,2579 | 0,0883 |
| NDVI | 0,7774 | 0,8053 | 0,8249 | 0,6826 | 0,6712 | 0,7573 | 0,6899 | 0,7100 | 0,7956 | 0,8226 | 0,6712 | 0,8249 | 0,1537 | 0,0551 |
| SR (value/100) | 0,0887 | 0,0981 | 0,1078 | 0,0662 | 0,0612 | 0,0827 | 0,0560 | 0,0604 | 0,0913 | 0,1113 | 0,0560 | 0,1078 | 0,0518 | 0,0177 |
| TVI | 1,1299 | 1,1422 | 1,1510 | 1,0856 | 1,0805 | 1,1207 | 1,0906 | 1,0999 | 1,1381 | 1,1498 | 1,0805 | 1,1510 | 0,0705 | 0,0252 |
| wNDII | 0,6386 | 0,6532 | 0,6555 | 0,4976 | 0,4952 | 0,5386 | 0,4918 | 0,4683 | 0,5592 | 0,5791 | 0,4683 | 0,6555 | 0,1872 | 0,0710 |

**Average: Non-Disturbed Area**

| Index/Date | 20.07.1992 | 02.07.1994 | 30.08.2001 | 02.09.2005 | 19.07.2006 | 22.07.2007 | 28.08.2009 | 17.07.2011 | 07.08.2013 | 12.07.2015 | MIN | MAX | MAX-MIN | St. Deviation |
|---|---|---|---|---|---|---|---|---|---|---|---|---|---|---|
| FMI (value*10) | 0,0965 | 0,1164 | 0,1980 | 0,1344 | 0,1259 | 0,0932 | 0,1393 | 0,1245 | 0,1352 | 0,1326 | 0,0932 | 0,1980 | 0,1048 | 0,0273 |
| NDMI | 0,3807 | 0,4213 | 0,3770 | 0,3462 | 0,3695 | 0,3283 | 0,3773 | 0,3694 | 0,4015 | 0,4064 | 0,3283 | 0,4213 | 0,0930 | 0,0262 |
| NDVI | 0,8108 | 0,8396 | 0,8812 | 0,8322 | 0,8218 | 0,8228 | 0,8588 | 0,8361 | 0,8726 | 0,8735 | 0,8108 | 0,8812 | 0,0705 | 0,0227 |
| SR (value/100) | 0,0984 | 0,1196 | 0,1678 | 0,1183 | 0,1096 | 0,1069 | 0,1359 | 0,1175 | 0,1516 | 0,1524 | 0,0984 | 0,1678 | 0,0694 | 0,0212 |
| TVI | 1,1448 | 1,1573 | 1,1752 | 1,1540 | 1,1495 | 1,1500 | 1,1656 | 1,1558 | 1,1716 | 1,1719 | 1,1448 | 1,1752 | 0,0304 | 0,0098 |
| wNDII | 0,6319 | 0,6614 | 0,6325 | 0,6060 | 0,6046 | 0,5948 | 0,6294 | 0,6234 | 0,6469 | 0,6502 | 0,5948 | 0,6614 | 0,0666 | 0,0200 |

**Figure A1.** Statistical information about the vegetation indices values in the Low Tatras NP (Source: Own work).

*Statistical Results for the Localities in the Sumava National Park*

**Locality 6**

| Index/Date | 30.07.1994 | 10.08.1998 | 28.07.2002 | 10.08.2004 | 29.08.2005 | 15.07.2006 | 19.08.2007 | 25.08.2009 | 10.08.2010 | 03.08.2013 | 09.08.2015 | MIN | MAX | MAX-MIN | St. Deviation |
|---|---|---|---|---|---|---|---|---|---|---|---|---|---|---|---|
| FMI (value*10) | 0,1066 | 0,1071 | | 0,1436 | 0,0977 | 0,1262 | 0,0878 | | 0,0295 | 0,0322 | 0,0439 | 0,0295 | 0,1436 | 0,1141 | 0,0392 |
| NDMI | 0,3869 | 0,3872 | | 0,4207 | 0,3875 | 0,4134 | 0,3655 | | 0,0592 | 0,1864 | 0,2262 | 0,0592 | 0,4207 | 0,3615 | 0,1197 |
| NDVI | 0,7540 | 0,7344 | No Data | 0,7080 | 0,7188 | 0,7347 | 0,7303 | No Data | 0,5868 | 0,6986 | 0,7612 | 0,5868 | 0,7612 | 0,1744 | 0,0488 |
| SR (value/100) | 0,0715 | 0,0659 | (Clouds) | 0,0593 | 0,0622 | 0,0660 | 0,0647 | (Clouds) | 0,0402 | 0,0565 | 0,0740 | 0,0402 | 0,0740 | 0,0338 | 0,0093 |
| TVI | 1,1198 | 1,1110 | | 1,0990 | 1,1039 | 1,1111 | 1,1091 | | 1,0420 | 1,0948 | 1,1230 | 1,0420 | 1,1230 | 0,0810 | 0,0227 |
| wNDII | 0,6375 | 0,6375 | | 0,6612 | 0,6378 | 0,6561 | 0,6228 | | 0,3817 | 0,4890 | 0,5200 | 0,3817 | 0,6612 | 0,2795 | 0,0915 |

**Locality 7**

| Index/Date | 30.07.1994 | 10.08.1998 | 28.07.2002 | 10.08.2004 | 29.08.2005 | 15.07.2006 | 19.08.2007 | 25.08.2009 | 10.08.2010 | 03.08.2013 | 09.08.2015 | MIN | MAX | MAX-MIN | St. Deviation |
|---|---|---|---|---|---|---|---|---|---|---|---|---|---|---|---|
| FMI (value*10) | 0,1231 | 0,1290 | | 0,1336 | 0,1206 | 0,1215 | 0,0845 | 0,0661 | 0,0988 | 0,0433 | 0,0328 | 0,0328 | 0,1336 | 0,1008 | 0,0351 |
| NDMI | 0,4422 | 0,4122 | | 0,3859 | 0,3996 | 0,3791 | 0,3571 | 0,3262 | 0,3463 | 0,2202 | 0,1535 | 0,1535 | 0,4422 | 0,2887 | 0,0851 |
| NDVI | 0,7748 | 0,7773 | No Data | 0,7373 | 0,7828 | 0,7560 | 0,7392 | 0,6780 | 0,7502 | 0,7256 | 0,7153 | 0,6780 | 0,7828 | 0,1048 | 0,0305 |
| SR (value/100) | 0,0795 | 0,0811 | (Clouds) | 0,0671 | 0,0830 | 0,0748 | 0,0677 | 0,0527 | 0,0704 | 0,0638 | 0,0606 | 0,0527 | 0,0830 | 0,0303 | 0,0092 |
| TVI | 1,1290 | 1,1301 | | 1,1123 | 1,1326 | 1,1205 | 1,1131 | 1,0853 | 1,1181 | 1,1070 | 1,1024 | 1,0853 | 1,1326 | 0,0473 | 0,0137 |
| wNDII | 0,6757 | 0,6553 | | 0,6369 | 0,6466 | 0,6319 | 0,6165 | 0,5936 | 0,6087 | 0,5155 | 0,4630 | 0,4630 | 0,6757 | 0,2127 | 0,0628 |

**Locality 8**

| Index/Date | 30.07.1994 | 10.08.1998 | 28.07.2002 | 10.08.2004 | 29.08.2005 | 15.07.2006 | 19.08.2007 | 25.08.2009 | 10.08.2010 | 03.08.2013 | 09.08.2015 | MIN | MAX | MAX-MIN | St. Deviation |
|---|---|---|---|---|---|---|---|---|---|---|---|---|---|---|---|
| FMI (value*10) | 0,0836 | 0,0890 | 0,0793 | 0,0960 | 0,0850 | 0,0994 | 0,0845 | 0,0716 | 0,1004 | 0,0769 | 0,0916 | 0,0716 | 0,1004 | 0,0288 | 0,0088 |
| NDMI | 0,3445 | 0,3486 | 0,3594 | 0,3333 | 0,3678 | 0,3891 | 0,3818 | 0,3433 | 0,3671 | 0,3788 | 0,3809 | 0,3333 | 0,3891 | 0,0558 | 0,0178 |
| NDVI | 0,7271 | 0,7265 | 0,7051 | 0,6857 | 0,7484 | 0,7441 | 0,7651 | 0,7118 | 0,7659 | 0,7616 | 0,7843 | 0,6857 | 0,7843 | 0,0986 | 0,0288 |
| SR (value/100) | 0,0686 | 0,0671 | 0,0593 | 0,0556 | 0,0711 | 0,0694 | 0,0759 | 0,0606 | 0,0771 | 0,0741 | 0,0830 | 0,0556 | 0,0830 | 0,0274 | 0,0079 |
| TVI | 1,1072 | 1,1071 | 1,0976 | 1,0886 | 1,1172 | 1,1153 | 1,1247 | 1,1007 | 1,1250 | 1,1232 | 1,1332 | 1,0886 | 1,1332 | 0,0446 | 0,0130 |
| wNDII | 0,6044 | 0,6094 | 0,6160 | 0,5979 | 0,6236 | 0,6388 | 0,6339 | 0,6061 | 0,6230 | 0,6316 | 0,6333 | 0,5979 | 0,6388 | 0,0409 | 0,0133 |

**Locality 9**

| Index/Date | 30.07.1994 | 10.08.1998 | 28.07.2002 | 10.08.2004 | 29.08.2005 | 15.07.2006 | 19.08.2007 | 25.08.2009 | 10.08.2010 | 03.08.2013 | 09.08.2015 | MIN | MAX | MAX-MIN | St. Deviation |
|---|---|---|---|---|---|---|---|---|---|---|---|---|---|---|---|
| FMI (value*10) | 0,1634 | 0,1510 | | 0,1460 | 0,1453 | 0,1475 | 0,1283 | 0,0940 | 0,1378 | 0,0328 | 0,0234 | 0,0234 | 0,1634 | 0,1400 | 0,0478 |
| NDMI | 0,4942 | 0,4655 | | 0,4365 | 0,4475 | 0,4728 | 0,4235 | 0,4398 | 0,4307 | 0,0959 | 0,0227 | 0,0227 | 0,4942 | 0,4715 | 0,1589 |
| NDVI | 0,7685 | 0,7523 | No Data | 0,7210 | 0,7726 | 0,7440 | 0,7653 | 0,6843 | 0,7514 | 0,6297 | 0,6164 | 0,6164 | 0,7726 | 0,1562 | 0,0547 |
| SR (value/100) | 0,0777 | 0,0713 | (Clouds) | 0,0628 | 0,0786 | 0,0684 | 0,0764 | 0,0536 | 0,0714 | 0,0442 | 0,0422 | 0,0422 | 0,0786 | 0,0364 | 0,0129 |
| TVI | 1,1262 | 1,1190 | | 1,1049 | 1,1281 | 1,1153 | 1,1248 | 1,0882 | 1,1186 | 1,0628 | 1,0566 | 1,0566 | 1,1281 | 0,0715 | 0,0250 |
| wNDII | 0,7104 | 0,6914 | | 0,6717 | 0,6793 | 0,6963 | 0,6631 | 0,6742 | 0,6680 | 0,4158 | 0,3533 | 0,3533 | 0,7104 | 0,3571 | 0,1205 |

**Locality 10**

| Index/Date | 30.07.1994 | 10.08.1998 | 28.07.2002 | 10.08.2004 | 29.08.2005 | 15.07.2006 | 19.08.2007 | 25.08.2009 | 10.08.2010 | 03.08.2013 | 09.08.2015 | MIN | MAX | MAX-MIN | St. Deviation |
|---|---|---|---|---|---|---|---|---|---|---|---|---|---|---|---|
| FMI (value*10) | 0,1666 | 0,1891 | | 0,2135 | 0,2059 | 0,2394 | 0,1662 | | 0,2187 | 0,1359 | 0,1666 | 0,1359 | 0,2394 | 0,1035 | 0,0310 |
| NDMI | 0,5500 | 0,5388 | | 0,5658 | 0,5518 | 0,5582 | 0,5129 | | 0,5394 | 0,5067 | 0,4902 | 0,4902 | 0,5658 | 0,0756 | 0,0243 |
| NDVI | 0,7985 | 0,8018 | No Data | 0,7664 | 0,8126 | 0,8085 | 0,7930 | No Data | 0,8021 | 0,7771 | 0,8001 | 0,7664 | 0,8126 | 0,0462 | 0,0140 |
| SR (value/100) | 0,0899 | 0,0918 | (Clouds) | 0,0766 | 0,0978 | 0,0967 | 0,0873 | (Clouds) | 0,0912 | 0,0799 | 0,0901 | 0,0766 | 0,0978 | 0,0212 | 0,0066 |
| TVI | 1,1395 | 1,1409 | | 1,1253 | 1,1456 | 1,1438 | 1,1371 | | 1,1411 | 1,1301 | 1,1402 | 1,1253 | 1,1456 | 0,0203 | 0,0062 |
| wNDII | 0,7464 | 0,7393 | | 0,7564 | 0,7474 | 0,7516 | 0,7226 | | 0,7397 | 0,7186 | 0,7078 | 0,7078 | 0,7564 | 0,0486 | 0,0156 |

**Average: Disturbed Area**

| Index/Date | 30.07.1994 | 10.08.1998 | 28.07.2002 | 10.08.2004 | 29.08.2005 | 15.07.2006 | 19.08.2007 | 25.08.2009 | 10.08.2010 | 03.08.2013 | 09.08.2015 | MIN | MAX | MAX-MIN | St. Deviation |
|---|---|---|---|---|---|---|---|---|---|---|---|---|---|---|---|
| FMI (value*10) | 0,1310 | 0,1290 | | 0,1411 | 0,1212 | 0,1317 | 0,1002 | 0,0801 | 0,0887 | 0,0361 | 0,0334 | 0,0334 | 0,1411 | 0,1077 | 0,0374 |
| NDMI | 0,4411 | 0,4216 | | 0,4144 | 0,4115 | 0,4218 | 0,3820 | 0,3830 | 0,2787 | 0,1675 | 0,1341 | 0,1341 | 0,4411 | 0,3070 | 0,1065 |
| NDVI | 0,7658 | 0,7547 | No Data | 0,7221 | 0,7581 | 0,7449 | 0,7449 | 0,6812 | 0,6961 | 0,6846 | 0,6976 | 0,6812 | 0,7658 | 0,0846 | 0,0309 |
| SR (value/100) | 0,0762 | 0,0728 | (Clouds) | 0,0631 | 0,0746 | 0,0696 | 0,0696 | 0,0532 | 0,0607 | 0,0548 | 0,0589 | 0,0532 | 0,0762 | 0,0231 | 0,0079 |
| TVI | 1,1250 | 1,1200 | | 1,1054 | 1,1215 | 1,1156 | 1,1157 | 1,0868 | 1,0929 | 1,0882 | 1,0940 | 1,0868 | 1,1250 | 0,0383 | 0,0141 |
| wNDII | 0,6745 | 0,6614 | | 0,6566 | 0,6546 | 0,6614 | 0,6341 | 0,6339 | 0,5528 | 0,4734 | 0,4454 | 0,4454 | 0,6745 | 0,2291 | 0,0797 |

**Average: Non-Disturbed Area**

| Index/Date | 30.07.1994 | 10.08.1998 | 28.07.2002 | 10.08.2004 | 29.08.2005 | 15.07.2006 | 19.08.2007 | 25.08.2009 | 10.08.2010 | 03.08.2013 | 09.08.2015 | MIN | MAX | MAX-MIN | St. Deviation |
|---|---|---|---|---|---|---|---|---|---|---|---|---|---|---|---|
| FMI (value*10) | 0,1251 | 0,1391 | | 0,1548 | 0,1455 | 0,1694 | 0,1254 | | 0,1596 | 0,1064 | 0,1291 | 0,0716 | 0,1694 | 0,0978 | 0,0300 |
| NDMI | 0,4473 | 0,4437 | | 0,4496 | 0,4598 | 0,4737 | 0,4474 | | 0,4533 | 0,4428 | 0,4356 | 0,3433 | 0,4737 | 0,1304 | 0,0395 |
| NDVI | 0,7628 | 0,7642 | No Data | 0,7261 | 0,7805 | 0,7763 | 0,7791 | No Data | 0,7840 | 0,7694 | 0,7922 | 0,7051 | 0,7922 | 0,0871 | 0,0290 |
| SR (value/100) | 0,0793 | 0,0795 | (Clouds) | 0,0661 | 0,0845 | 0,0831 | 0,0816 | (Clouds) | 0,0842 | 0,0770 | 0,0866 | 0,0593 | 0,0866 | 0,0273 | 0,0094 |
| TVI | 1,1234 | 1,1240 | | 1,1070 | 1,1314 | 1,1296 | 1,1309 | | 1,1331 | 1,1267 | 1,1367 | 1,0976 | 1,1367 | 0,0391 | 0,0130 |
| wNDII | 0,6754 | 0,6744 | | 0,6772 | 0,6855 | 0,6952 | 0,6783 | | 0,6814 | 0,6751 | 0,6706 | 0,6061 | 0,6952 | 0,0891 | 0,0271 |

**Figure A2.** Statistical information about the vegetation indices values in the Sumava NP (Source: Own work).

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
