# Peer review of "Evaluation of the Influence of Disturbances on Forest Vegetation Using the Time Series of Landsat Data: A Comparison Study of the Low Tatras and Sumava National Parks"

_ijgi, doi:10.3390/ijgi8020071_

Round 1

Reviewer 1 Report

General comments:

The authors present a work evaluating the forest vegetation changes of 6 locations from 1992 to 2015 in the Low Tatras National Park in Slovakia and Sumava National park in the Czechia using the time series (TS) of Landsat images. The language of the manuscript is in most cases appropriate but could benefit from additional proof-reading.

However, the contribution and innovation of this work is not clearly addressed. This manuscript lacks some details on the methods part (e.g. what is the criteria or method to evaluate the disturbance). I suggest the authors select more sites for different types of disturbance and calculate the average vegetation indices. In addition, it might be worthwhile to consider this work at spatial scale.

Detailed read-through comments:

1.       Line 19: Please provide full name of CDR and PIF here.

2.       Line 20: By “excellent abilities”, are there any indices or methods to measure the ability?

3.       Line 36: Due to the low temporal resolution (for example, unlike Sentinel-2) of Landsat image?

4.       Line 40: what does “data rows” mean?

5.       Line 43: remove “tree”

6.       Line 51: optic->medium; data of hyperspectral->hyperspectral data

7.       Line 52: This data helps-> These data help ...

8.       Line 54: helps->help

9.       Line 72: In my opinion, if the authors want to validate the results, another set of in-situ data should be used. So more than one site should be selected for each type of disturbance.  

10.   Line 92: It is about a-> It is an about

11.   Line 132: 3.1->2.1

12.   Line 133: Please provide the full name of CDR here.

13.   Line 138: add “between 1992 and 2015” after “11 images were finally selected”

14.   Line 139: 11 images selected-> 11 images were selected

15.   Line 140: An->Another

16.   Line 141: remove “also”

17.   Line 141-142: remove year (2012)…(2012,2014)

18.   Line 158: 3.1->2.2

19.   Line 163-164: How can the time series analyses be used for estimating spectral characteristics? Or based on the vegetation indices to do what? Please revise this sentence.

20.   Line 165: The Landsat data source here means the non-CDR data? What are the difference between the CDR and non-CDR data?

21.   Line 171: remove the second “used”

22.   Line 175: Please provide the full name of PIF the first time it appears in the paper.

23.   Line 176: “…the most relevant studies [33]…” Add more references.

24.   Line 179: “TimeSync web application”  Any references?

25.   Line 184: remove the sentence after “curves created.”

26.   Line 185: How did you consider the surrounding pixels when calculating vegetation indices and why?

27.   Line 186-187: remove “e.g., Chen et al. (2005), Rouse et al. (1973)…”

You don't need to list all the author's name. Just list the references.

28.   Line 190-207: It is better to use a table to list these vegetation indices.

Line 209-210: What is unmasking? what is the purpose of unmasking? I suggest remove this sentence. Remove “ArcMap was the software which was used to create maps.”

29.   Line 227: X axis? Could you provide more description on what you did to different vegetation indices and why? e.g. value*10 and value/100.

30.   Line 233-235: What's the difference between the two trends?

31.   Line 271, Figure 8: One image was acquired in May 2007 and the next one was acquired in Feb 2010. The difference in vegetation indices may be influenced by the difference in growing season. Did you consider this factor?

32.   Line 296: “…of this study is obvious…”->  “…of this study, it is obvious…”

33.   Line 299: It is not clear by “a large impact decrease”

34.   Line 300-301: Why a forestry intervention will cause an increase in the reflection in the green and near infrared part of spectrum.

35.   Line 313-314: ...played an important role in what? Please make this sentence clearer.

36.   Line 315-318: This sentence is too long. Please revise it.

37.   Line 319: Landsat data-> Landsat imagery

38.   Lind 326: Reference

Author Response

Dear reviwer,

We thank you for your helpful and inspiring comments. With regard to the pieces of advice, we significantly modified essential parts of the manuscript (Abstract, Methodology, Results, Discussion and Conclusion and also, we added Attachment part).

We believe that our improvements make the manuscript better and more interesting for a wide range of readers.

Mgr. Josef Lastovicka

Department of Applied Geoinformatics and Cartography,

Faculty of Science, Charles University

On behalf of all authors.

General comments:

The authors present a work evaluating the forest vegetation changes of 6 locations from 1992 to 2015 in the Low Tatras National Park in Slovakia and Sumava National park in the Czechia using the time series (TS) of Landsat images. The language of the manuscript is in most cases appropriate but could benefit from additional proof-reading.

However, the contribution and innovation of this work is not clearly addressed. This manuscript lacks some details on the methods part (e.g. what is the criteria or method to evaluate the disturbance). I suggest the authors select more sites for different types of disturbance and calculate the average vegetation indices. In addition, it might be worthwhile to consider this work at spatial scale.

Response for General comments: Thank you very much for your factual comments. Following the suggestion, we added in the method part a detailed description of PIF Linear Based method that we used. We added more observed localities as well. We calculated average vegetation indices for localities with and without disturbations for both study areas. Then we added into the results part new figures with statistical calculation/results for individual localities and for both types of localities (with and without disturbations) as well. Moreover, we improved the discussion part and we created the new attachment with results of all the calculation.

Detailed read-through comments:

1.       Line 19: Please provide full name of CDR and PIF here.

Response 1: Thank you for the comment, we added full names for both shortcuts.

2.       Line 20: By “excellent abilities”, are there any indices or methods to measure the ability?

Response 2: Thank you for the comment, it was a mistake.  We corrected it.

3.       Line 36: Due to the low temporal resolution (for example, unlike Sentinel-2) of Landsat image?

Response 3: Yes, exactly, of Landsat image, we completed the sentence and now it is more understandable.

4.       Line 40: what does “data rows” mean?

Response 4: Thank you for the comment, it was a mistake: “Raw data (like DN)”.

5.       Line 43: remove “tree”

Response 5: Removed.

6.       Line 51: optic->medium; data of hyperspectral->hyperspectral data

Response 6: Changed.

7.       Line 52: This data helps-> These data help ...

Response 7: Thank you, we corrected it.

8.       Line 54: helps->help

Response 8: Thank you, we corrected it.

9.       Line 72: In my opinion, if the authors want to validate the results, another set of in-situ data should be used. So more than one site should be selected for each type of disturbance.

Response 9: Thank you, we added four new localities (one locality with and one locality without disturbance in both areas of interest).

10.   Line 92: It is about a-> It is an about

Response 10: Thank you, we corrected it. (We used an, but our English corrector change it to “in”.)

11.   Line 132: 3.1->2.1

Response 11: Thank you, we corrected it.

12.   Line 133: Please provide the full name of CDR here.

Response 12: Yes, we provided the full name.

13.   Line 138: add “between 1992 and 2015” after “11 images were finally selected”

Response 13: Thank you, we added it.

14.   Line 139: 11 images selected-> 11 images were selected

Response 14: Thanks, we changed it.

15.   Line 140: An->Another

Response 15: Changed, thanks.

16.   Line 141: remove “also”

Response 16: Removed, thanks.

17.   Line 141-142: remove year (2012)(2012,2014)

Response 17: Removed, thanks.

18.   Line 158: 3.1->2.2

Response 18: Corrected, thanks.

19.   Line 163-164: How can the time series analyses be used for estimating spectral characteristics? Or based on the vegetation indices to do what? Please revise this sentence.

Response 19: Thank you very much, it was a mistake. We corrected it.

20.   Line 165: The Landsat data source here means the non-CDR data? What are the difference between the CDR and non-CDR data?

Response 20: CDR data means Climate Data Record, so CDR data are with Level 2 correction.

21.   Line 171: remove the second “used”

Response 21: Removed, thanks.

22.   Line 175: Please provide the full name of PIF the first time it appears in the paper.

Response 22: Yes, thank you, we corrected it.

23.   Line 176: “…the most relevant studies [33]” Add more references.

Response 23: Yes, thank you, we added all new part with many references.

24.   Line 179: “TimeSync web application” Any references?

Response 24: We added reference, university name and also the URL link.

25.   Line 184: remove the sentence after “curves created.”

Response 25: Thank you, we corrected it.

26.   Line 185: How did you consider the surrounding pixels when calculating vegetation indices and why?

Response 26: We used all pixels in the neighbourhood of central pixel, where GPS point was found. The value of index was calculated as an average of these 9 values. Reason: we did not use any geodetic, professional GPS, but a standard GPS receiver. Using this method, we inhibited a lower accuracy of our GPS measurement.

27.   Line 186-187: remove “e.g., Chen et al. (2005), Rouse et al. (1973)” You don't need to list all the author's name. Just list the references.

Response 27: Names were removed.

28.   Line 190-207: It is better to use a table to list these vegetation indices.

Response 28: Yes, thank you, we created a list.

29.  Line 209-210: What is unmasking? what is the purpose of unmasking? I suggest remove this sentence. Remove “ArcMap was the software which was used to create maps.”

Response 29: Yes, thank you, we added information about Fmask unmasking and also about methods. We used Fmask algorithm for unmasking clouds, shadows, snow etc. We removed this part: “ArcMap was the software which was used to create maps.”

30.   Line 227: X axis? Could you provide more description on what you did to different vegetation indices and why? e.g. value*10 and value/100.

Response 30: Yes, thank you very much for this comment. We added more detailed information into the methods part.

31.   Line 233-235: What's the difference between the two trends?

Response 31: Yes, thank you. Now it is corrected and more understandable.

32.   Line 271, Figure 8: One image was acquired in May 2007 and the next one was acquired in Feb 2010. The difference in vegetation indices may be influenced by the difference in growing season. Did you consider this factor?

Response 32: Yes, this is really good question, question of a comparability. Especially, in mountain areas, there are not so many cloud free satellite images. We were forced to use images from different months. On the other hand,  this problem should eliminates RRN methods. To make these images comparable we used PIF Linear Based normalization (one of the RRN methods). From this point of view, we also updated the methods part to be more understandable this problem. We were referring to our article from year 2017 (Lastovicka et al.), where we tested CDR data with and without normalizations with in-situ measurements. On this article PIF Linear Based normalization was chosen.

33.   Line 296: “…of this study is obvious…”->  “…of this study, it is obvious…”

Response 33: Thank you, we corrected it.

34.   Line 299: It is not clear by “a large impact decrease”

Response 34: Yes, thanks, we completed the sentence with the part in brackets: “This corresponds to a large impact decrease (more than with bark beetle calamity, because biotic calamity is slower decreasing) in the indices using RED, NIR and SWIR bands [44]”

35.   Line 300-301: Why a forestry intervention will cause an increase in the reflection in the green and near infrared part of spectrum.

Response 35: Yes, thank you for this question. Explanation: after disturbance, foresters planted new trees to initiate/speed up “rehabilitation” of the forest. New trees together with successive vegetation reflected in GREEN and NIR bands, there is more chlorophyll than during the calamity (damaged trees) and new cell structure, which can increase of reflectance in GREEN and NIR band. Reference: Jensen, J. Remote Sensing of the Environment: An Earth Resource Perspective. Pearson Prentice Hall. 2007. University of Minnesota. 592 p. ISBN: 0131889508.

36.   Line 313-314: ...played an important role in what? Please make this sentence clearer.

Response 36: Yes, thank you very much, we completed the sentence: “It is clear from the results that this aspect played an important role in detecting disturbed period (compare the NDMI or wNDII vegetation indices used SWIR band with the vegetation indices based on NIR like NDVI)”

37.   Line 315-318: This sentence is too long. Please revise it.

Response 37: We revised it, now there are two sentences.

38.   Line 319: Landsat data-> Landsat imagery

Response 38: Thanks, we changed it.

39.   Line 326: Reference

Response 39: Thank you very much, it was a mistake. We removed this sentence.

Reviewer 2 Report

This work studied on monitoring forest vegetation changes due to wind damage and bark beetle attack using the time series of Landsat images. The study is clear and easily understands, but the method and analysis is too simple and lack of new ideas. The conclusion is well known, which implied that RS technique is a useful tool for detecting forest changes. Therefore, I suggest rejecting this work in the present form. Some comments on this work are as follows:

1.     The sample sites using in this study is too few. Only six sites were used, which are not enough to support your results. For example, the suitable vegetation indices for detecting forest changes were presented in this works is not valid due to limited sample sites. I suggest using more sites and add mean and standard deviation values in Figures.

2.     The PIF method using in this work is not clear. The author should present how to select PIF according to your study area.

3.     Some quantitative methods are suggested to be used for capturing disturbance and recovery points. Such as LandTrendr-Temporal segmentation algorithms (Remote Sensing of Environment 114 (2010) 2897–2910)

4.     The discussion should be more concrete.

Author Response

Dear reviwer,

We thank you for your helpful and inspiring comments. With regard to the pieces of advice, we significantly modified essential parts of the manuscript (Abstract, Methodology, Results, Discussion and Conclusion and also, we added Attachment part).

We believe that our improvements make the manuscript better and more interesting for a wide range of readers.

Mgr. Josef Lastovicka

Department of Applied Geoinformatics and Cartography,

Faculty of Science, Charles University

On behalf of all authors.

Response to Reviewer 2 Comments

This work studied on monitoring forest vegetation changes due to wind damage and bark beetle attack using the time series of Landsat images. The study is clear and easily understands, but the method and analysis is too simple and lack of new ideas. The conclusion is well known, which implied that RS technique is a useful tool for detecting forest changes. Therefore, I suggest rejecting this work in the present form. Some comments on this work are as follows:

Response for General comments: Thank you very much for your factual comments. Following your suggestion, we added in the method part detailed information about PIF Linear Based method that we used. From method point of view, this article used and develop advanced methods of RRN. Methods were used for creating TS graphs, from this point of view this article brings new information about testing RRN methods in connection with TS analyses. In your comments, you mentioned LandTrendr, which could be alternative for our own app. Unfortunately, this algorithm does not use the normalizations. We consider this a factor that could influenced in negative way the statistical data, especially when it comes to comparison of two measurements from different times/months of the year. The environment is not homogenous during the long-term period of observation. If the normalization is not used, the results could be affected by this phenomenon. So that was the reason to use the normalizations for our datasets, not the LandTrendr app disposal in the time of our analysis. We decided to created our own app in MATLAB for creating time series figures and also for half-automatic normalization using PIF Linear Based. Nowadays we are also testing alternative normalization methods. We introduced our app and a study about normalizations in already published article (Lastovicka et al. 2017).

On the other hand, new version of the LandTrendr algorithm was published in last weeks, it is called LandsatLinkr (http://jdbcode.github.io/LandsatLinkr/ or https://landsat.usgs.gov/landsatlinkr-harmonizing-landsat-archive or https://www.sciencedirect.com/science/article/pii/S2352340917305474 or https://www.sciencedirect.com/science/article/abs/pii/S0034425718300579). This new algorithm is alternative for our own app, because there is a limited set of tools of the normalizations focussing on different types of sensor. But there is used normalization only between MSS and ETM/TM, and between ETM+ and OLI. The normalization between MSS /TM/ETM and ETM+/OLI is missing. Nowadays we are preparing new article focused on normalizations methods (PIF and IR-MAD) comparing with LandsatLinkr TC analyses. So many thanks for your very good comments, we consider it as very relevant.

1.     The sample sites using in this study is too few. Only six sites were used, which are not enough to support your results. For example, the suitable vegetation indices for detecting forest changes were presented in this works is not valid due to limited sample sites. I suggest using more sites and add mean and standard deviation values in Figures.

Response 1: Thank you very much for your valuable notes and correction suggestions. We added 4 more sites into our study. In conclusion or study contains 10 localities and we also created new graphs with statistical information about all localities. There I also included maximum, minimum, difference of maximal and minimal values, and standard deviation. This approach was really useful for interpretation of our figures in general.

2.     The PIF method using in this work is not clear. The author should present how to select PIF according to your study area.

Response 2: Thank you for the comment, we added more detailed information into the methodology part. We included a more detailed explanation of our methods. Now all the steps of processing the data should be more understandable in this article.

3.     Some quantitative methods are suggested to be used for capturing disturbance and recovery points. Such as LandTrendr-Temporal segmentation algorithms (Remote Sensing of Environment 114 (2010) 2897–2910)

Response 3: Thank you very much for your comment. We added quantitative methods allowing comparison of our results (Statistical approach), exactly minimum, maximum, difference of maximum and minimum and standard deviation as well. Then we calculated average values of each study area (Low Tatras and Sumava) for both types of localities – with disturbances and with low level of disturbances.

4.     The discussion should be more concrete.

Response 4: Thank you very much. We improved the conclusion and discussion parts. We focused on deeper explanation of our results, wider discussion about the used methods and an application of our results and methods in the management of protected areas.

Round 2

Reviewer 1 Report

General comments:

The authors made significant revisions to the manuscript by adding more study sites and the average and statistical information. The paper can be published after some minor revisions.

Detailed read-through comments:

1.       Line 143-144: “For the Low Tatras NP, 11 images were finally selected (see Table 3) and, for the Sumava NP, 11 images were selected between 1992 and 2015 (see Table 4)”->”11 images were finally selected for the Low Tatras NP (Table 3) and the Sumava NP (Table 4) respectively.”

2.       Line 176: “Fmask feature…”  Reference

3.       Line 176: “unmask”-> I think “mask” should be used in this context

4.       Line 203: remove . right after “cluster”.

5.       Line 235: “unmasking”-> “masking”   

6.       Line 235: “calculated”->”performed”

7.       Line 240: remove “ in the software MS Excel”

8.       Line 259: Figure 5-Figure 18: 11 images were selected for each site but why there are only 9 or 10 values instead of 11 in the figures? The authors should explain it. The same for the following figures.

Please add a main X axis title (here must be Date) for all the figures. Also, please mark the date of each image clearly. The readers cannot tell the exact date of each image in graphs presenting like this.

Add (a) and (b) for the two graphs instead of just describe them as upper/bottom graph

The figures need to be improved by changing the text font, color, outline, graph size and other format. It is lengthy and not professional by just listing the 14 figures presenting in such format.

9.       Line 268-269: “These results can be also seen from the statistical information in the bottom graph, which was calculated for all the sites.”

How? What is the meaning of the bottom graph? Can the authors explain it? (same for the following figures)

10.   Line 311: As you mentioned before, since "there is a bark beetle calamity, culminating in 2009 with a peak occurring in 2012". How does this affect the time series vegetation indices? I did see any decrease in the VI time series during 2009 and 2012. Please give some description and discussion.

Author Response

Dear reviewer,

We thank you for the review and for helpful and inspiring comments. We tried to make revision as carefully as we could, and we used almost all of their comments and recommendations. We modified all graphs and we corrected all suggestions.

We believe that our improvements make the manuscript better. 

Mgr. Josef Lastovicka

Department of Applied Geoinformatics and Cartography, Faculty of Science, Charles University

Reviewer 1

General comments:

The authors made significant revisions to the manuscript by adding more study sites and the average and statistical information. The paper can be published after some minor revisions.

Response for General comments: Thank you very much for your answer for the corrections of our article. And also we thank you for your comments. We corrected all of them.

Detailed read-through comments:

1.       Line 143-144: “For the Low Tatras NP, 11 images were finally selected (see Table 3) and, for the Sumava NP, 11 images were selected between 1992 and 2015 (see Table 4)”->”11 images were finally selected for the Low Tatras NP (Table 3) and the Sumava NP (Table 4) respectively.”

Response 1: Thank you, corrected.

2.       Line 176:“Fmask feature…”  Reference

Response 2: Yes, we added the reference.

3.       Line 176:“unmask”-> I think “mask” should be used in this context

Response 3: Thank you, we changed it.

4.       Line 203: remove “.” right after “cluster”.

Response 4: Removed.

5.       Line 235:“unmasking”-> “masking”   

Response 5: Changed.

6.       Line 235:“calculated”->”performed”

Response 6: Changed.

7.       Line 240:remove “ in the software MS Excel”

Response 7: Thanks, removed.

8.       Line 259:Figure 5-Figure 18: 11 images were selected for each site but why there are only 9 or 10 values instead of 11 in the figures? The authors should explain it. The same for the following figures.

Response 8a:Yes, thank you very much for this comment. We added an explanation for it (lines 276 – 279). This problem was caused by a cloud cover of image/images. We selected the best 11 images covering all the sites with minimal cloud cover. Although we did this selection, it was not ensured no cloud infection for all the sites for all the observed years. In some cases, there was a cloud in the pixels. In this case, we used NaN value, we left this point and we used linear interpolation between nearest points. More details see Fig. 19 – 20 in the attachment. 

Please add a main X axis title (here must be Date) for all the figures. Also, please mark the date of each image clearly. The readers cannot tell the exact date of each image in graphs presenting like this.

Response 8b:X axis titles added, dates were changed.

Add (a) and (b) for the two graphs instead of just describe them as upper/bottom graph

Response 8c:We added (a)/(b). 

The figures need to be improved by changing the text font, color, outline, graph size and other format. It is lengthy and not professional by just listing the 14 figures presenting in such format.

Response 8d:All figures were changed, resized, we used same font as in text (also with black color).

9.       Line 268-269:“These results can be also seen from the statistical information in the bottom graph, which was calculated for all the sites.”

How? What is the meaning of the bottom graph? Can the authors explain it? (same for the following figures)

Response 9: We explained that in text and also we changed words “bottom graph” to Figure (b).

10.   Line 311:As you mentioned before, since "there is a bark beetle calamity, culminating in 2009 with a peak occurring in 2012". How does this affect the time series vegetation indices? I did see any decrease in the VI time series during 2009 and 2012. Please give some description and discussion.

Response 10:Yes, thank you for this comment. We have specified time of disturbances more preciously for our observed/selected sites, especially for the observed locality 4. In the locality 4, wind calamity occurred in 2004. In the graph 8, we can see effect of this calamity next year, because the calamity was in autumn. Then/in the following months, the bark beetle calamity expanded to this locality. On the other hand, the bark beetle calamity occurred later in the locality 2. The biggest enlargement of this calamity (culmination) occurred 2006 – 2009. Out of our selected sites, the bark beetle calamity culminated after 2009 (2009 – 2012). We deleted this note about bark beetle calamity 2009-2012 from our text, because this process was occurred out of our selected areas. You are right, this fact was not well specified in our article, so we clarified it more precisely, see lines 96 – 102. Thank you very much for the alert.

Reviewer 2 Report

The authors have carefully revised the manuscript according to the reviewers' comments, thus I agree to publish this paper on IJRS. Congratulations.

Author Response

Dear reviewer,

We thank you for the review. We tried to make new revision as carefully as we could. We modified all graphs and few sentences.

We believe that our improvements make the manuscript better. 

Mgr. Josef Lastovicka

Department of Applied Geoinformatics and Cartography, Faculty of Science, Charles University
